# Model Correlation Detection via Random Selection Probing

## Abstract

The growing prevalence of large language models (LLMs) and vision–language models (VLMs) has heightened the need for reliable techniques to determine whether a model has been fine-tuned from or is even identical to another. Existing similarity-based methods often require access to model parameters or produce heuristic scores without principled thresholds, limiting their applicability. We introduce Random Selection Probing (RSP), a hypothesis-testing framework that formulates model correlation detection as a statistical test. RSP optimizes textual or visual prefixes on a reference model for a random selection task and evaluates their transferability to a target model, producing rigorous p-values that quantify evidence of correlation. To mitigate false positives, RSP incorporates an unrelated baseline model to filter out generic, transferable features. We evaluate RSP across both LLMs and VLMs under diverse access conditions for reference models and test models. Experiments on fine-tuned and open-source models show that RSP consistently yields small p-values for related models while maintaining high p-values for unrelated ones. Extensive ablation studies further demonstrate the robustness of RSP. These results establish RSP as the first principled and general statistical framework for model correlation detection, enabling transparent and interpretable decisions in modern machine learning ecosystems.

## 1 Introduction

The rapid proliferation of large language models (AI@Meta, 2024; Team, 2024) and vision-language models (Team, 2025b) has created an urgent need for reliable methods to determine whether a given model has been fine-tuned from another or is even identical. Such detection is critical for ensuring transparency, accountability, and intellectual property protection in modern machine learning ecosystems. As models are increasingly shared, adapted, and redeployed, the ability to establish lineage is essential not only for research reproducibility but also for legal and ethical considerations.

Existing approaches to model similarity can be categorized into representational and functional measures (Klabunde et al., 2025). Representational methods compare internal activations to quantify similarity (Raghu et al., 2017; Kornblith et al., 2019), while functional methods operate on outputs, employing metrics such as disagreement rates (Madani et al., 2004) or divergence (Lin, 2002). Despite their usefulness, these approaches face two critical limitations. First, many require access to model parameters, architectures, or intermediate activations, which is an unrealistic assumption in the case of proprietary systems. Second, they typically produce heuristic similarity scores without principled thresholds, leaving ambiguity about whether two models are truly correlated.

To overcome these limitations, we introduce **Random Selection Probing** (RSP), a statistical framework that formulates model correlation detection as a hypothesis test. Rather than producing heuristic similarity scores, our method outputs statistically rigorous $p$-values, quantifying the evidence of correlation between a reference and a target model. RSP operates by optimizing textual or visual prefixes on the reference model for a random selection task, e.g., "randomly choose a character from a to z", to maximize the probability of producing a specific token. The transferability of these optimized prefixes is then evaluated on the test model. To further reduce false positives, we incorporate an unrelated baseline model that prevents the generation of generic, transferable prefixes.

Our experimental results on finetuned and open source models demonstrate that RSP is effective across both LLMs and VLMs, and under diverse accessibility conditions. For reference models, RSP operates under gradient-accessible and logits-accessible settings. For test models, it supports

both gray-box settings, where logits are available, and black-box settings, where only output text is observed. Across all scenarios, RSP consistently produces very small $p$-values for related models while avoiding false positives on unrelated ones, highlighting both the robustness and generality of our approach.

Our contributions are summarized as follows:

- We propose the first principled hypothesis-testing framework for model correlation detection, providing statistically rigorous $p$-values that enable clear and interpretable decisions.
- We introduce a novel random selection probing task and design optimization methods for both LLMs and VLMs under diverse access conditions.
- We conduct extensive experiments on different models and settings, showing that RSP reliably identifies correlations on finetuned and related open source models, while avoiding false positives on unrelated models.

## 2 RELATED WORK

### 2.1 MODEL SIMILARITY

A growing body of work has investigated methods for quantifying similarity between neural network models. Broadly, these approaches can be divided into *representational* and *functional* similarity measures (Klabunde et al., 2025). Representational similarity focuses on comparing intermediate activations, with techniques such as canonical correlation analysis (CCA), centered kernel alignment (CKA), and Procrustes-based metrics (Raghu et al., 2017; Kornblith et al., 2019). These methods reveal how internal representations align across models, but they may not directly capture functional behavior.

Functional similarity measures, in contrast, operate on model outputs. Performance-based and prediction-based metrics include disagreement rates (Madani et al., 2004), Jensen–Shannon divergence (Lin, 2002), and surrogate churn (Klabunde et al., 2025). More fine-grained approaches leverage gradients or adversarial perturbations, such as ModelDiff (Li et al., 2021), and saliency map similarity (Jones et al., 2022). Stitching-based methods further assess compatibility by training small adapters between models and evaluating downstream performance (Bansal et al., 2021).

Existing approaches suffer from two primary limitations. First, many of them require access to model weights, which is infeasible in the case of proprietary models. Second, they typically yield only a similarity score, for which it is nontrivial to determine an appropriate threshold. In contrast, the proposed RSP produces a $p$-value, thereby providing a statistically principled criterion for assessing whether two models are correlated.

### 2.2 ADVERSARIAL ATTACK

Our work builds upon adversarial attack methods to optimize model prefixes. In the white-box setting, where gradients are accessible, projected gradient descent (PGD) (Madry et al., 2018) has emerged as a standard baseline for generating robust adversarial examples by iteratively updating perturbations under norm constraints. More recent developments, such as Gradient-based Combinatorial Generation (GCG) (Zou et al., 2023), adapt gradient information to optimize universal adversarial prompts for language models, demonstrating strong transferability across tasks. Auto-DAN (Zhu et al.) further automates the generation of adversarial natural language instructions by integrating large language models into the optimization loop.

In black-box settings, where gradients are unavailable, alternative strategies are required. Zeroth-Order Optimization (ZOO) (Chen et al., 2017) estimates gradients through finite-difference methods, enabling adversarial perturbation even without model internals. Bandit-based approaches (Ilyas et al., 2018) reduce query complexity by exploiting gradient priors, making black-box adversarial attacks significantly more efficient.

### 2.3 MODEL FINGERPRINT

Our method is also close to the concept of model fingerprint. Xu et al. (2024) introduce Instructional Fingerprinting, which implants secret key–response pairs through lightweight instruction tuning to ensure persistence under fine-tuning. Russinovich & Salem (2024) propose Chain & Hash, a cryptographic method that binds prompts and responses to provide verifiable, unforgeable ownership.

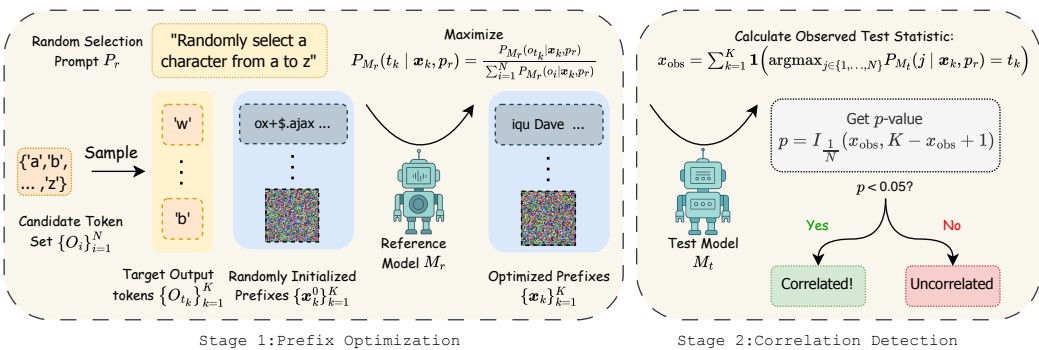

Figure 1: Overview of the Random Selection Probing (RSP) framework. RSP operates in two stages: (1) **Prefix Optimization**, where textual or visual prefixes are optimized on the reference model for a random selection task, and (2) **Correlation Detection**, where the transferability of the optimized prefixes is evaluated on the test model. The resulting statistical test produces a $p$-value, enabling principled detection of model correlations.

Pasquini et al. (2025) develop LLMmap, an active fingerprinting technique that identifies model versions via crafted queries, enabling accurate recognition across varied deployment settings. Zhang et al. (2024) present REEF, a training-free method that uses representation similarity to detect model derivations robustly under fine-tuning, pruning, and permutation.

## 3 RANDOM SELECTION PROBING

This section introduces the proposed RSP framework. We begin with a high-level overview, followed by the algorithms tailored to different model families and experimental settings. As illustrated in Figure 1, RSP operates in two stages. **Stage 1: Prefix Optimization.** A collection of prefixes is optimized on a designated reference model $M_r$. **Stage 2: Correlation Detection.** The statistical correlation between the reference model $M_r$ and a target model $M_t$ is assessed by testing whether the optimized prefixes preserve their effectiveness when transferred from $M_r$ to $M_t$.

### 3.1 PREFIX OPTIMIZATION

---

**Algorithm 1** Prefix Optimization Procedure

---

**Require:** Reference model $M_r$, random selection prompt $p_r$, random initializations of prefixes $\{x_k^0\}_{k=1}^K$, target output tokens $\{o_{t_k}\}_{k=1}^K$, prefix optimization function $f$, maximum update rounds $R_{\max}$.

1: **for** $k \leftarrow 1$ to $K$ **do**
2:     Initialize $x_k \leftarrow x_k^0$
3:     **for** $R \leftarrow 1$ to $R_{\max}$ **do**
4:         $x_k \leftarrow f(M_r, p_r, x_k, o_{t_k})$
5:     **end for**
6: **end for**
7: **return** $\{x_k\}_{k=1}^K$

---

To quantify transferability, we formulate a *random selection probing* task. Concretely, the reference model $M_r$, either a VLM or a LLM, is prompted with a random selection prompt $p_r$, which requires the model to choose a token uniformly from a candidate output set $\{o_i\}_{i=1}^N$, where $o_i \in V$ for $i = 1, \ldots, N$, $V$ is the vocabulary. The objective is to optimize a collection of textual or visual prefixes $\{x_k\}_{k=1}^K$, initialized as $\{x_k^0\}_{k=1}^K$, such that each prefix $x_k$ maximizes the probability of its designated target token $o_{t_k}$, with $t_k \in \{1, \ldots, N\}$. Formally, for each $k$ we maximize

$$P_{M_r}(t_k \mid x_k, p_r) := \frac{P_{M_r}(o_{t_k} \mid x_k, p_r)}{\sum_{i=1}^N P_{M_r}(o_i \mid x_k, p_r)}. \tag{1}$$

The complete optimization procedure is provided in Algorithm 1. In the following sections, we describe the implementation of the prefix optimization function $f$ across different settings.

## 3.2 Textual Prefix Optimization in Large Language Models

A textual prefix that induces a high probability of generating a desired random token can be decomposed into two types of features: **model-specific features** and **general features**. General features correspond to the semantic content of the prefix, which universally increases the likelihood of the target token across different LLMs. For example, the prefix "*output letter c*" can directly bias multiple models toward generating the token "c." In contrast, model-specific features exploit idiosyncratic patterns unique to a given model family, and thus do not readily transfer to unrelated models.

In our setting, the objective is to optimize prefixes that function exclusively within the reference model family while exhibiting minimal transferability to unrelated models. That is, the optimized prefix should primarily encode model-specific features, while suppressing general features. To enforce this constraint, we introduce an unrelated model $M_u$ and require that, during optimization, the probability of generating the target token under $M_u$ is minimized.

---

**Algorithm 2** Optimization function $f$ for LLMs with gradients

---

**Require:** reference model $M_r$, unrelated model $M_u$, random selection prompt $p_r$, vocabulary $V$, input textual prefix $\boldsymbol{x} \in V^L$, target token index $t$
    Initialize candidate pool $\mathcal{X} \leftarrow \{\}$
    Encode $\boldsymbol{x}$ into one-hot matrix $E \in \{0,1\}^{L \times |V|}$
    **for** $i$ in $1, \cdots, L$ **do**
        Get Top-k replacements $\mathcal{X}_i$ from $-\nabla_{E_i} \log P_{M_r}(t \mid \boldsymbol{x}, p_r)$ based on GCG Zou et al. (2023).
        $\mathcal{X} \leftarrow \mathcal{X} \cup \mathcal{X}_i$
    **end for**
    $\boldsymbol{x} \leftarrow \arg\max_{\boldsymbol{x}' \in \mathcal{X}} (P_{M_r}(t \mid \boldsymbol{x}', p_r) - P_{M_u}(t \mid \boldsymbol{x}', p_r))$
    **return** $\boldsymbol{x}$

---

**Gradient Access.** When gradients are available for the reference model, we adopt a gradient-guided search approach inspired by Greedy Coordinate Gradient (GCG) (Zou et al., 2023). In GCG, tokens are updated iteratively: at each step, a candidate token is greedily selected from a replacement pool so as to minimize the model loss. The replacement pool consists of the top-$k$ tokens with the smallest gradients when represented in one-hot form. However, this procedure may inadvertently introduce general features, resulting in false positives across unrelated models. To mitigate this issue, we modify the optimization objective by incorporating $M_u$. Specifically, instead of maximizing only $P_{M_r}(t \mid \boldsymbol{x}, p_r)$, we maximize the difference $P_{M_r}(t \mid \boldsymbol{x}, p_r) - P_{M_u}(t \mid \boldsymbol{x}, p_r)$, thereby encouraging model-specific rather than general features. The detailed optimization procedure for $f$ is presented in Alg. 2.

---

**Algorithm 3** Optimization function $f$ for LLMs with logits

---

**Require:** reference model $M_r$, unrelated model $M_u$, random selection prompt $p_r$, word list $V$, input textual prefix $\boldsymbol{x} \in V^L$, target output token index $t$, number of mutations $B_{\text{LLM}}$, mutation probability $p_{\text{mutate}}$
    Initialize candidate pool $\mathcal{X} \leftarrow \{\}$
    **for** $i$ in $1, \cdots, B_{\text{LLM}}$ **do**
        $\boldsymbol{x}^i \leftarrow \boldsymbol{x}$
        **for** $j$ in $1, \cdots, L$ **do**
            Replace $\boldsymbol{x}^i_j$ with another random word with the probability $p_{\text{mutate}}$
        **end for**
        $\mathcal{X} \leftarrow \mathcal{X} \cup \{\boldsymbol{x}^i\}$
    **end for**
    $\boldsymbol{x} \leftarrow \arg\max_{\boldsymbol{x}' \in \mathcal{X}} (P_{M_r}(t \mid \boldsymbol{x}', p_r) - P_{M_u}(t \mid \boldsymbol{x}', p_r))$
    **return** $\boldsymbol{x}$

---

**Logit Access.** When only the logits or output probabilities of the reference model are available, we adopt a genetic-algorithm-inspired search strategy for the random selection task. In this setting, the prefix $\boldsymbol{x}$ is treated as a sequence of words, since we assume no access to the tokenizer. At each iteration, we generate $B_{\text{LLM}}$ candidate mutations by randomly replacing words in $\boldsymbol{x}$ with probability $p_{\text{mutate}}$. Among these candidates, we retain the one that maximizes $P_{M_r}(t \mid \boldsymbol{x}, p_r) - P_{M_u}(t \mid \boldsymbol{x}, p_r)$, as outlined in Alg. 3.

### 3.3 VISUAL PREFIX OPTIMIZATION IN VISION–LANGUAGE MODELS

For vision-language models, it is particularly challenging to generate transferable visual patterns from randomly initialized noise. Consequently, we do not require an additional unrelated model $M_u$ in this setting.

**Gradient Access.** When gradients are accessible, we directly adopt projected gradient descent (PGD) (Madry et al., 2018) to optimize the visual prefix. Given a visual prefix $\boldsymbol{x} \in \mathbb{Z}_{256}^{H \times W \times 3}$, the prefix optimization function $f$ is defined as

$$f_{\text{VLM}}^{\text{grad}}(M_r, p_r, \boldsymbol{x}) \;=\; \text{clip}\left(\boldsymbol{x} - \text{sgn}(-\nabla_{\boldsymbol{x}} \log P_{M_r}(t \mid \boldsymbol{x}, p_r)), \, 0, \, 255\right), \tag{2}$$

where sgn denotes the sign function and $t$ is the target output token index.

**Logits Access.** Although genetic algorithms are effective for optimizing textual prefixes, we find them less suitable for VLMs. Instead, a more natural approach is to adopt a zeroth-order optimization method to estimate the gradient required by PGD. Specifically, for the visual prefix $\boldsymbol{x} \in \mathbb{Z}_{256}^{H \times W \times 3}$, we draw $B_{\text{VLM}}$ random perturbation vectors $u_i \in \{+1, -1\}^{H \times W \times 3}$ and construct perturbed samples

$$\boldsymbol{x}_1^i = \text{clip}(\boldsymbol{x} + u_i, 0, 255), \quad \boldsymbol{x}_2^i = \text{clip}(\boldsymbol{x} - u_i, 0, 255). \tag{3}$$

We then approximate the gradient via a symmetric finite difference:

$$\hat{\nabla}_{\boldsymbol{x}} P_{M_r}(t \mid \boldsymbol{x}, p_r) = \frac{1}{B_{\text{VLM}}} \sum_{i=1}^{B_{\text{VLM}}} \frac{\log P_{M_r}(t \mid \boldsymbol{x}_1^i, p_r) - \log P_{M_r}(t \mid \boldsymbol{x}_2^i, p_r)}{\boldsymbol{x}_1^i - \boldsymbol{x}_2^i}. \tag{4}$$

The resulting estimate can then be substituted into Eq. 2 to iteratively optimize the visual prefix.

### 3.4 CORRELATION DETECTION

Given the optimized textual or visual prefix set $\{\boldsymbol{x}_k\}_{k=1}^K$, we evaluate their performance on the test model $M_t$ in order to assess the presence of statistical correlation between the reference model $M_r$ and $M_t$. Two evaluation scenarios are considered: the *gray-box* setting and the *black-box* setting. In the gray-box setting, neither the architecture nor the parameters of $M_t$ are accessible; however, query access to output logits or top-$k$ log-probabilities is available, as is the case for proprietary systems such as GPT-4 and Gemini. In the black-box setting, only the generated output text is observable.

**Gray-Box.** Correlation is evaluated through a hypothesis test. The null hypothesis is defined as $H_0$: $M_t$ and $M_r$ are independent, such that optimized prefixes obtained from $M_r$ do not transfer to $M_t$. The alternative hypothesis is $H_1$: $M_t$ and $M_r$ exhibit correlation, such that optimized prefixes successfully transfer. To this end, we consider the predictive distribution $P_{M_t}(t \mid \boldsymbol{x}, p_r)$. Under $H_0$, the optimized prefixes are not transferable. Let $X$ denote the number of prefixes for which the designated target token attains the highest probability. Then $X$ follows a binomial distribution, i.e., $X \sim B\left(K, \frac{1}{N}\right)$. Note that the test model may exhibit inherent biases toward certain choices. However, this does not affect the validity of our hypothesis test, because the target token is uniformly sampled from the candidate set. A detailed proof is provided in Appendix H.1. The observed test statistic is given by

$$x_{\text{obs}} = \sum_{k=1}^K \mathbf{1}\left(\underset{j \in \{1, \ldots, N\}}{\arg \max} \, P_{M_t}(j \mid \boldsymbol{x}_k, p_r) = t_k\right). \tag{5}$$

The corresponding $p$-value can then be expressed as

$$p = \Pr(X \geq x_{\text{obs}}) = I_{\frac{1}{N}}(x_{\text{obs}}, K - x_{\text{obs}} + 1), \tag{6}$$

where $I_x(a, b) = \frac{B(x; a, b)}{B(a, b)}$ denotes the regularized incomplete beta function, with $B(x; a, b)$ and $B(a, b)$ denoting the incomplete and complete beta functions, respectively. A $p$-value less than the significance threshold of 0.05 constitutes statistical evidence to reject $H_0$ in favor of $H_1$, thereby supporting the presence of correlation between $M_t$ and $M_r$.

Table 1: Model correlation detection $p$-values on finetuned LLMs. Our proposed RSP, achieves $p$-values below the 0.05 threshold across both gradient-access and logits-access settings for $M_r$, under both gray-box and black-box conditions for $M_t$.

| | | Gray-Box | | | Black-Box | | |
|---|---|---|---|---|---|---|---|
| | | GSM8K | Dolly-15k | Alpaca | GSM8K | Dolly-15k | Alpaca |
| Grad | Llama-8B | 9.08e-240 | 1.48e-4 | 1.17e-6 | 3.12e-225 | 1.47e-5 | 6.49e-6 |
| | Qwen2.5-3B | 7.31e-57 | 6.02e-4 | 5.62e-13 | 9.40e-51 | 7.05e-5 | 1.06e-8 |
| | Phi-4-mini | 1.00e-300 | 1.54e-79 | 4.34e-177 | 1.00e-300 | 1.17e-72 | 5.39e-154 |
| Logits | Llama-8B | 1.00e-300 | 7.43e-9 | 1.80e-11 | 1.00e-300 | 1.37e-9 | 5.77e-12 |
| | Qwen2.5-3B | 1.18e-254 | 3.99e-3 | 2.02e-2 | 1.13e-257 | 6.02e-4 | 1.21e-2 |
| | Phi-4-mini | 1.00e-300 | 4.31e-118 | 7.65e-163 | 1.00e-300 | 9.08e-106 | 4.66e-132 |

Table 2: Model correlation detection $p$-value results on finetuned VLMs. Visual prefix optimization with PGD is more effective than optimizing textual prefixes, yield very small $p$-values.

| | Gray-Box | | Black-Box | |
|---|---|---|---|---|
| | Visual7w | MathV360k | Visual7w | MathV360k |
| Qwen2.5-VL-7B | 1.00e-300 | 3.02e-208 | 1.00e-300 | 1.83e-205 |
| Llama-3.2-11B-Vision | 1.00e-300 | 1.14e-226 | 1.00e-300 | 6.13e-221 |

**Black-Box.** In the black-box setting, where only text outputs are observable, the probability-maximizing token in Eq. 5 cannot be accessed directly. To approximate this quantity, we query the model $T$ times and estimate the most probable token via empirical frequency. The resulting counts are then substituted into Eq. 6 to compute the corresponding $p$-value.

## 4 EXPERIMENTS

In this section, we present the experimental results of our proposed method, RSP. The detailed experimental settings and hyperparameters are provided in Appendix C, while additional experiments are reported in Appendix E.

### 4.1 MODELS AND DATASETS

We evaluate our method across diverse models and datasets. For LLM experiments, we adopt Llama-3-8B-Instruct (AI@Meta, 2024), Qwen2.5-3B-Instruct (Team, 2024), and Phi-4-mini-instruct (Abouelenin et al., 2025) as reference models $M_r$, and fine-tune them on GSM8k (Cobbe et al., 2021), Dolly-15k (Conover et al., 2023), and Alpaca (Taori et al., 2023). For VLMs, we employ Qwen2.5-VL-7B-Instruct (Team, 2025b) and Llama-3.2-11B-Vision-Instruct (AI@Meta, 2024), fine-tuned on Visual7w (Zhu et al., 2016) and MathV360k (Shi et al., 2024). The details of the fine-tuning procedure and hyperparameter configurations are provided in Appendix B. In addition, we examine the correlations between the reference models and publicly released models fine-tuned from them.

### 4.2 RESULTS ON FINETUNED MODELS

The $p$-value results for model correlation detection are presented in Table 1 for LLMs and Table 2 for VLMs. Using a significance threshold of 0.05, our RSP consistently detects correlations between the reference model $M_r$ and the test model $M_t$ with high confidence. This holds across both LLMs and VLMs, regardless of whether gradient access or logits access is available for $M_r$, and under both gray-box and black-box settings for $M_t$. To account for the numerical limits of double-precision floating-point representation, we cap the minimum reportable $p$-value at $1.00 \times 10^{-300}$.

### 4.3 RESULTS ON OPEN SOURCE MODELS

We further evaluate our method on a range of open-source models, including those finetuned from Llama-3-8B-Instruct and Qwen2.5-VL-7B-Instruct backbones. As shown in Table 3, our approach consistently produces small $p$-values when detecting correlations between Llama-3-8B-Instruct and

Table 3: Model correlation detection $p$-values between Llama-3-8B-Instruct and other open-source models. The results demonstrate that our method effectively captures correlations between the reference and test models, even after large-scale finetuning.

|  | Grad | | Logits | |
|---|---|---|---|---|
|  | Gray-Box | Black-Box | Gray-Box | Black-Box |
| Llama-3.1-8B-Instruct (AI@Meta, 2024) | 1.70e-13 | 4.78e-10 | 1.50e-82 | 6.21e-67 |
| Llama-3.2-3B-Instruct (AI@Meta, 2024) | 1.48e-4 | 3.03e-4 | 1.48e-14 | 6.23e-18 |
| Bio-Medical-Llama-3-8B (Con, 2024) | 3.03e-4 | 1.16e-3 | 1.67e-27 | 1.67e-27 |
| Llama-3.1-Swallow-8B (Okazaki et al., 2024) | 1.16e-3 | 3.03e-4 | 7.41e-41 | 1.19e-41 |
| llama-3-Korean-Bllossom-8B (Choi et al., 2024) | 4.63e-65 | 7.31e-57 | 4.12e-228 | 4.82e-211 |
| Llama-3-Instruct-8B-SimPO-v0.2 (Meng et al., 2024) | 5.65e-108 | 7.20e-107 | 7.38e-172 | 8.92e-176 |

Table 4: Model correlation detection results on open-source models finetuned from Qwen2.5-VL-7B-Instruct. The results show that our method identifies strong correlations with very high confidence.

|  | Grad | | Logits | |
|---|---|---|---|---|
|  | Gray-Box | Black-Box | Gray-Box | Black-Box |
| VLAA-Thinker-Qwen2.5VL-7B (Chen et al., 2025) | 1.00e-300 | 1.00e-300 | 3.29e-16 | 1.61e-18 |
| ThinkLite-VL-7B (Wang et al., 2025) | 1.00e-300 | 1.00e-300 | 1.70e-13 | 1.19e-15 |
| Qwen2.5-VL-7B-Instruct-abliterated (huihui-ai, 2025) | 1.00e-300 | 1.00e-300 | 1.48e-14 | 1.70e-13 |
| qwen2.5-vl-7b-cam-motion-preview (Lin et al., 2025) | 1.00e-300 | 1.00e-300 | 1.64e-10 | 3.27e-5 |

its derivatives, confirming that the learned prefixes successfully transfer even after large-scale finetuning across diverse domains and languages. Similarly, for models finetuned from Qwen2.5-VL-7B-Instruct in Table 4, our method yields small $p$-values across both gray-box and black-box settings, highlighting its robustness and sensitivity. These results provide strong evidence that our statistical test can reliably identify lineage relationships among open-source models, demonstrating high confidence in correlation detection across different architectures and finetuning strategies.

## 4.4 CASE STUDY

Table 5 presents two examples of optimized textual prefixes. While these prefixes do not convey any interpretable semantic meaning to humans, they consistently induce the model to generate the designated target token. Because optimized visual prefixes appear indistinguishable from random noise to human observers, they are omitted from the main text. Additional examples of both visual and textual prefixes are provided in Appendix G.

## 5 ANALYSIS

### 5.1 ABLATION STUDY

In this section, we analyze the effects of different hyperparameters. Additional ablation results for prefix length $L$ and mutation probability $p_{mutate}$ are provided in Appendix D.

**Number of Samples.** As shown in Figure 2, increasing the number of samples consistently reduces the $p$-value, with a clear trend across both gray-box and black-box settings for LLMs and

Table 5: Textual prefixes optimized with Qwen2.5-3B-Instruct.

|  | Textual Prefixes | Target Output Token |
|---|---|---|
| Grad | Official-firstanut dernugePP Poker Circ amenk dc national mobil relig threat MLmdl \u0142yreadcrumbs_opts{ prevHETxtypressipelineContinue browsces InputStream[pLoadingCurrencystheft stamp useStyles NPCtbl):\r\nEHRFwrite ImageSun findsitialHistor CHEath | n |
| Logits | samplers $842,617 McNeil tab-lifter 139-foot clothbound freeze-out insecticide indictment kidding terrier hovering Allotments articulate Linus 126,000 fiendish diplomats Estimate Fromm 4,369 railbirds shipboard years unequally share-holders beef-hungry Mercers Pinkie conformance flapped Indians' annex anxiety hello Apprehensively 160,000 hens' inventories Counseling address Boaz Marsha silly concedes neat hooting 42 Moisture Ambassador-designate | h |

Table 6: $p$-value results across different resolutions on Qwen2.5-VL-7B-instruct. Lower resolutions (e.g., $140\times140$) may not provide sufficient information, whereas higher resolutions (e.g., $560\times560$) increase the difficulty of optimization.

| Model | Resolution | Gray-Box | | Black-Box | |
|---|---|---|---|---|---|
| | | Visual7w | MathV360k | Visual7w | MathV360k |
| Qwen2.5-VL-7B | $140\times140$ | **1.00e-300** | 1.56e-141 | 4.22e-144 | 5.39e-66 |
| | $280\times280$ | **1.00e-300** | 3.02e-208 | **1.00e-300** | **1.83e-205** |
| | $560\times560$ | **1.00e-300** | **4.82e-211** | 3.02e-208 | 4.99e-159 |

Table 7: Correlation test results between Qwen2.5-3B-Instruct and other models. Without the unrelated model $M_u$ in Alg. 2 and Alg. 3, the optimized prefixes may occasionally yield false positives on models not closely related to Qwen2.5-3B-Instruct. Values below the significance threshold of 0.05 are underlined.

| | Grad | | | | Logits | | | |
|---|---|---|---|---|---|---|---|---|
| | Gray-Box | | Black-Box | | Gray-Box | | Black-Box | |
| | Ours | w/o $M_u$ | Ours | w/o $M_u$ | Ours | w/o $M_u$ | Ours | w/o $M_u$ |
| Llama-3-8B | 1.13e-1 | 8.67e-1 | 7.71e-2 | 8.67e-1 | 9.14e-1 | 9.93e-1 | 9.71e-1 | 9.85e-1 |
| Qwen3-4B | 7.30e-1 | 1.60e-1 | 8.67e-1 | 1.13e-1 | 2.19e-1 | 6.45e-1 | 2.19e-1 | 3.72e-1 |
| DeepSeek-R1-Qwen3-8B | 1.13e-1 | 8.05e-1 | 7.71e-2 | 8.05e-1 | 5.10e-1 | 5.54e-1 | 1.60e-1 | 4.61e-1 |
| DeepSeek-R1-Llama-8B | 3.72e-1 | 9.48e-1 | 3.72e-1 | 8.67e-1 | 7.30e-1 | 2.19e-1 | 7.30e-1 | 1.60e-1 |
| Mistral-7B | 3.72e-1 | 3.99e-3 | 3.72e-1 | 3.26e-2 | 1.60e-1 | 1.47e-5 | 2.90e-1 | 5.48e-11 |

VLMs. These results confirm that larger sample sizes substantially enhance the statistical power of our method, making correlation detection more reliable. We also provide the results on unrelated models in Figure 4. The results show that unrelated models consistently yield large $p$-values. However, no clear trend is observed, as the $p$-values for unrelated models are largely affected by randomness.

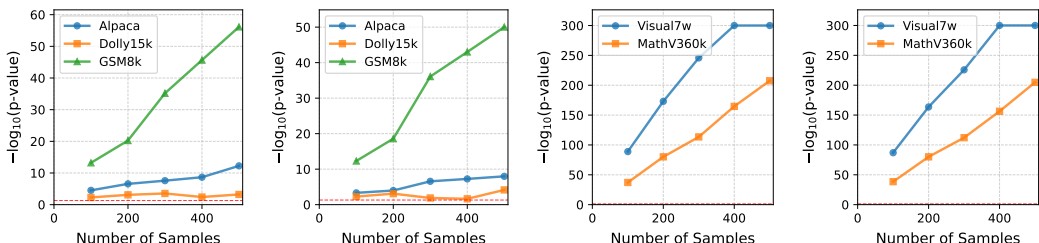

(a) Gray-box $p$-values on Qwen2.5-3B-Instruct. (b) Black-box $p$-values on Qwen2.5-3B-Instruct. (c) Gray-box $p$-values on Qwen2.5-VL-7B-Instruct (d) Black-box $p$-values on Qwen2.5-VL-7B-Instruct

Figure 2: Ablation study on the number of samples. Increasing the number of samples consistently reduces the resulting $p$-value. The red dotted line denotes the significance threshold at 0.05.

**Mutation Probability.** We further investigate the effect of the mutation probability $p_{\text{mutate}}$ on correlation detection. As illustrated in Figure 3, the influence of $p_{\text{mutate}}$ varies across datasets, and a range of values can be effective. Notably, even when setting $p_{\text{mutate}} = 1$, i.e., generating a completely new prefix for each mutation, the method is still able to identify a prefix that successfully fulfills the task.

**Prefix Length.** We perform an ablation study on prefix length using Qwen2.5-3B-Instruct to assess the robustness of RSP. As shown in Table 14, the method remains effective even with very short prefixes of only 10 tokens, yielding extremely small $p$-values under both gray-box and black-box settings. In addition, shorter prefixes tend to produce smaller $p$-values than longer ones, indicating that compact representations are sufficient to capture model correlations with high statistical significance. However, shorter prefixes more easily violate the independence assumption required for

Table 8: Model correlation detection $p$-values on unrelated models. We evaluate Qwen2.5-VL-7B-Instruct, Llama-3.2-11B-Vision-Instruct, llava-v1.6-mistral-7b-hf (Liu et al., 2023), and gemma-3-4b-it (Team, 2025a). The consistently high $p$-values indicate that the optimized prefixes do not transfer to unrelated models, thereby preventing false positives.

| | Gray-Box | | | | Black-Box | | | |
|---|---|---|---|---|---|---|---|---|
| | Qwen2.5-VL | Llama3.2-V | LLaVa-1.6 | Gemma 3 | Qwen2.5-VL | Llama3.2-V | LLaVa-1.6 | Gemma 3 |
| Qwen2.5-VL | - | 0.290 | 0.290 | 0.971 | - | 0.290 | 0.461 | 0.805 |
| Llama-3.2-V | 0.290 | - | 0.290 | 0.971 | 0.290 | - | 0.461 | 0.805 |

Table 9: Textual prefix similarity across different prefix lengths. We evaluate the Qwen2.5-3B-Instruct model under the gradient access setting. Prefixes are encoded into embeddings using Sentence-BERT, and cosine similarity is computed to measure their representational similarity.

| | Average Similarity↓ | | | Top 1% Similarity↓ | | |
|---|---|---|---|---|---|---|
| Prefix Length | 10 | 20 | 50 | 10 | 20 | 50 |
| Random Prefixes | **0.1327** | **0.1897** | 0.3053 | **0.3220** | **0.3687** | **0.4654** |
| RSP | 0.1390 | 0.1926 | **0.2973** | 0.3454 | 0.3779 | 0.4660 |

the statistical test, as they occasionally generate identical tokens or words, as shown in Table 15. To mitigate this issue, we adopt a longer prefix length of 50 in our main experiments, where we do not observe such collisions. A more detailed analysis of the independence of optimized prefixes is provided in Sec. 5.3.

**Resolution.** We also examine the effect of input resolution on correlation detection. As presented in Table 6, lower resolutions like $140{\times}140$ may not contain sufficient information for reliable detection, while very high resolutions, e.g., $560{\times}560$ introduce additional optimization challenges. The intermediate resolution of $280{\times}280$ provides a favorable balance, yielding consistently strong performance across both gray-box and black-box settings.

## 5.2 UNRELATED MODELS

We evaluate the correlation between reference LLMs, VLMs, and other models in Table 7 and Table 8. The results demonstrate that our method does not yield false positives, as unrelated models consistently produce large $p$-values. For LLMs, we show that incorporating the unrelated model $M_u$ in Alg. 2 and Alg. 3 is effective and necessary in mitigating the generation of transferable prefixes. Without $M_u$, the optimization process may occasionally produce prefixes with general features, which can inadvertently lead to false positives.

## 5.3 INDEPENDENCE ANALAYSIS

The validity of our $p$-values relies on the assumption that the generated textual or visual prefixes are independent. To validate the $p$-value calculation in Eq. 6, we assess whether the optimized prefixes exhibit sufficient independence.

For LLMs, we employ Sentence-BERT (Reimers & Gurevych, 2019) to encode the textual prefixes into embeddings and compute both the average cosine similarity and the top 1% cosine similarity. As reported in Table 9, when the prefix length is set to 50, the similarity among optimized prefixes is nearly indistinguishable from that of randomly generated prefixes. However, for shorter lengths, e.g., 10 and 20, the similarity is slightly higher than the random baseline.

For VLMs, we directly compute cosine similarity using pixel values, with results summarized in Table 18. These results indicate that the visual prefixes are substantially diverse, which is expected given that the parameter space of visual prefixes is much larger than that of text.

## 6 Conclusion

We introduced Random Selection Probing (RSP), a statistical framework for model correlation detection that provides rigorous p-values rather than heuristic similarity scores. By optimizing prefixes on a reference model and testing their transferability to a target model, RSP reliably detects lineage across LLMs and VLMs under diverse settings. Experiments on fine-tuned and open-source models show that RSP achieves extremely small p-values for related models while avoiding false positives on unrelated ones. These results establish RSP as a robust and general tool for transparent model auditing, with promising extensions to broader multimodal and security applications.

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

## A    USAGE OF LLMs

LLMs are used to polish and assist in writing the paper.

## B    TRAINING DETAILS

Table 10: **Qwen2.5-3B-Instruct fine-tuning hperparameters.**

|  | alpaca_cleaned | dolly15k_alpaca | gsm8k_alpaca |
|---|---|---|---|
| Batch Size | 64 | 64 | 64 |
| Epochs / Steps | 3 epochs | 3 epochs | 100 steps |
| LR | $2 \times 10^{-4}$ | $2 \times 10^{-4}$ | $1 \times 10^{-5}$ |
| Warmup | 0.03 | 0.05 | 0.15 |
| Weight Decay | 0.01 | 0.01 | 0.05 |
| LoRA (r/$\alpha$/drop) | 16/32/0.05 | 16/32/0.05 | 16/16/0.05 |
| MaxLen | 2048 | 2048 | 3072 |
| Pack | on | on | off |

Table 11: **Llama3-8B-Instruct fine-tuning hperparameters.**

|  | alpaca_cleaned | dolly15k_alpaca | gsm8k_alpaca |
|---|---|---|---|
| B$\times$A | $4 \times 4$ | $4 \times 4$ | $4 \times 4$ |
| Epochs / Steps | 3 epochs | 4 epochs | 100 steps |
| LR | $2 \times 10^{-4}$ | $2 \times 10^{-4}$ | $1 \times 10^{-5}$ |
| Warmup | 0.03 | 0.03 | 0.15 |
| Weight Decay | 0.01 | 0.01 | 0.05 |
| LoRA (r/$\alpha$/drop) | 16/32/0.05 | 16/32/0.05 | 16/16/0.05 |
| MaxLen | 2048 | 2048 | 3072 |
| Pack | on | on | off |

Table 12: **Phi-4-mini-Instruct fine-tuning hperparameters.**

|  | alpaca_cleaned | dolly15k_alpaca | gsm8k_alpaca |
|---|---|---|---|
| B$\times$A | $2 \times 8$ | $4 \times 4$ | $4 \times 4$ |
| Epochs / Steps | 3 epochs | 4 epochs | 100 steps |
| LR | $2 \times 10^{-4}$ | $2 \times 10^{-4}$ | $1 \times 10^{-5}$ |
| Warmup | 0.03 | 0.03 | 0.15 |
| Weight Decay | 0.01 | 0.01 | 0.05 |
| LoRA (r/$\alpha$/drop) | 16/32/0.05 | 16/32/0.05 | 16/16/0.05 |
| MaxLen | 2048 | 2048 | 3072 |
| Pack | on | on | off |

The training parameters for LLMs are presented in Tables 10, 11, 12, and these for VLMs are presented in Table 13.

## C    HYPERPARAMETERS

In our experiments, we use the random selection prompt $p_r$ = "Randomly choose a letter from a to z. Only output the chosen letter in your response with nothing else." The corresponding candidate output set consists of the 26 English letters, i.e., $N = 26$. We generate $K = 500$ prefixes in total. For textual prefixes, the sequence length is fixed at 50 tokens in the gradient-access setting and 50 words in the logits-access setting. For image prefixes, we adopt images of resolution $280 \times 280$, i.e., $H = W = 280$. In the logits-access setting, the number of candidate mutations for both LLMs and VLMs is set to $B_{\text{LLM}} = B_{\text{VLM}} = 32$. The query time $T$ for black-box settings is set to 100. The maximum number of optimization rounds is set to 100 for the gradient-access setting and 1000 for the logits-access setting. For the unrelated model $M_u$, we employ Phi-4-mini-instruct in the experiments with Llama-3-8B-Instruct and Qwen2.5-3B-Instruct. Conversely, in the

Table 13: **VL models fine-tuning hyperparameters** for Qwen2.5-VL-7B-Instruct and Llama-3.2-11B-Vision-Instruct on two datasets.

| Parameter | Qwen2.5-VL-7B-Instruct | | Llama-3.2-11B-Vision-Instruct | |
|---|---|---|---|---|
| | MathV360k | Visual7w | MathV360k | Visual7w |
| Batch Size | 64 | 64 | 64 | 64 |
| Epochs / Steps | 3 epochs | 2 epochs | 3 epochs | 2 epochs |
| LR | $8 \times 10^{-5}$ | $5 \times 10^{-5}$ | $8 \times 10^{-5}$ | $5 \times 10^{-5}$ |
| Warmup | 0.03 | 0.05 | 0.03 | 0.05 |
| Weight Decay | 0.01 | 0.01 | 0.01 | 0.01 |
| LoRA (r/$\alpha$/drop) | 16/32/0.05 | 8/16/0.05 | 16/32/0.05 | 8/16/0.05 |
| LoRA Target | all | q_proj,v_proj | all | q_proj,v_proj |

experiments with Phi-4-mini-instruct as the reference model, we use Qwen2.5-3B-Instruct as $M_u$. The experiments are run on 8 NVIDIA H100 GPUs.

# D  ABLATION STUDY

## D.1  PREFIX LENGTH

Table 14: Ablation results on prefix length. We test it on Qwen2.5-3B-Instruct. The results show that RSP works with even only 10 tokens, and short prefixes produce smaller $p$-values.

| Model | Prefix Length | Gray-Box | | | Black-Box | | |
|---|---|---|---|---|---|---|---|
| | | GSM8K | Dolly-15k | Alpaca | GSM8K | Dolly-15k | Alpaca |
| Qwen2.5-3B | 10 | **1.49e-146** | 1.12e-2 | **9.64e-32** | **1.24e-135** | 6.02e-4 | **1.72e-23** |
| | 20 | 1.09e-88 | **3.27e-5** | 1.70e-13 | 7.33e-93 | **1.17e-6** | 5.61e-13 |
| | 50 | 7.31e-57 | 6.02e-4 | 5.62e-13 | 9.40e-51 | 7.05e-5 | 1.06e-8 |

We perform an ablation study on prefix length using Qwen2.5-3B-Instruct to assess the robustness of RSP. As shown in Table 14, the method remains effective even with very short prefixes of only 10 tokens, yielding extremely small $p$-values under both gray-box and black-box settings.

In addition, shorter prefixes tend to produce smaller $p$-values than longer ones, indicating that compact representations are sufficient to capture model correlations with high statistical significance. However, shorter prefixes more easily violate the independence assumption required for the statistical test, as they occasionally generate identical tokens or words, as shown in Table 15.

To mitigate this issue, we adopt a longer prefix length of 50 in our main experiments, where we do not observe such collisions. A more detailed analysis of the independence of optimized prefixes is provided in Sec. 5.3.

## D.2  MUTATION PROBABILITY.

We further investigate the effect of the mutation probability $p_{\text{mutate}}$ on correlation detection. As illustrated in Figure 3, the influence of $p_{\text{mutate}}$ varies across datasets, and a range of values can be

Table 15: Prefix collisions in short prefixes, where prefix length is set to 10. Identical tokens are highlighted in **bold**. Such collisions may violate the independence assumption required for the statistical test. To avoid this issue, we set the prefix length to 50 in our experiments.

| Textual Prefixes | Target Output Token |
|---|---|
| DiplgpDoc smssenal.**ISupportInitialize** Instance.Err_summaryylon BE\u00b0 intelligenceSn\u0632.**ISupportInitialize** MERCHANTABILITY governance storageyon | y |
| *MPDF **migrationBuilder/**apache cle.reload fuel —— enabledOTE **migrationBuilder**.intellij experimentinar*) | o |

Table 16: Correlation detection results on Llama3.2-1B-Instruct finetuned on OpenOrca. Our method consistently identifies strong correlations under both gray-box and black-box settings, with extremely small $p$-values across gradient-based and logit-based analyses.

|  | Gray-Box | Black-Box |
|---|---|---|
| Grad | 1.26e-86 | 1.46e-83 |
| Logits | 1.37e-9 | 4.79e-10 |

Table 17: $p$-value results for Qwen2.5-VL-7B-Instruct under logits access. Values below the 0.05 significance threshold are highlighted in **bold**.

|  |  | Gray-Box | Black-Box |
|---|---|---|---|
| Finetuned Model | Visual7w | **4.77e-7** | **7.43e-8** |
|  | MathV360k | **2.02e-2** | **1.16e-3** |
| Unrelated Model | Llama-3.2-11B-Vision-Instruct | 2.19e-1 | 2.90e-1 |
|  | llava-v1.6-mistral-7b-hf | 7.71e-2 | 7.71e-2 |
|  | gemma-3-4b-it | 5.54e-1 | 8.05e-11 |

effective. Notably, even when setting $p_{\text{mutate}} = 1$, i.e., generating a completely new prefix for each mutation, the method is still able to identify a prefix that successfully fulfills the task.

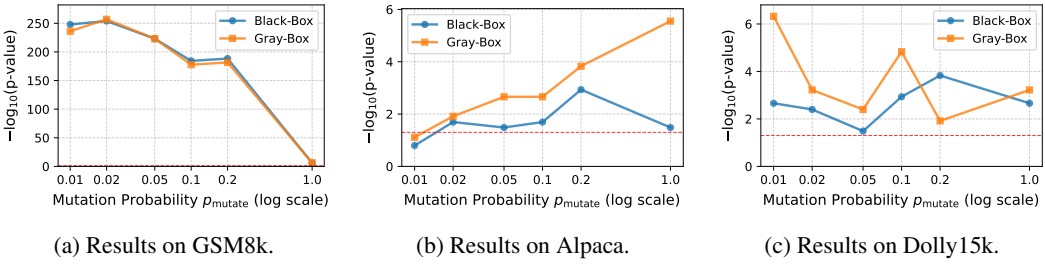

(a) Results on GSM8k.  (b) Results on Alpaca.  (c) Results on Dolly15k.

Figure 3: Ablation results for different mutation probability $p_{\text{mutate}}$.

# E  ADDITIONAL EXPERIMENTS

## E.1  RESULTS ON OPENORCA

To evaluate whether our method can still detect correlations after extensive finetuning, we applied it to Llama3.2-1B-Instruct finetuned on OpenOrca (Lian et al., 2023), which contains approximately 3M training samples. The results, presented in Table 16, demonstrate that our method continues to effectively capture the correlation, yielding an extremely small $p$-value.

## E.2  RESULTS FOR QWEN2.5-VL-7B-INSTRUCT WITH LOGITS ACCESS.

We provide additional results for Qwen2.5-VL-7B-instruct under logits access in Table 17.

## E.3  INDEPENDENCE ANALYSIS FOR VLMS

Table 18 demonstrates that the visual prefixes are highly diverse, exhibiting extremely low similarity.

Table 18: Similarity results for VLMs. Using optimized prefixes obtained from Qwen2.5-VL-7B-Instruct, we compute cosine similarities directly from pixel values. The results indicate that the optimized visual prefixes exhibit substantial diversity.

| | Average Similarity↓ | Top 1% Similarity↓ |
|---|---|---|
| Grad | 7.74e-5 | 8.38e-3 |
| Logits | -2.85e-6 | 8.27e-3 |

Table 19: Time efficiency analysis for RSP. Here $k$ is the number of Top-K choices in GCG.

| | | Forward | Backward | Total Time Cost |
|---|---|---|---|---|
| LLM | Grad | $2R_{\max}Lk = 30000$ | $R_{\max} = 100$ | $\sim 65$ s |
| | Logits | $2R_{\max}B_{\text{LLM}} = 64000$ | 0 | $\sim 3$ min |
| VLM | Grad | $R_{\max} = 100$ | $R_{\max} = 100$ | $\sim 23$ s |
| | Logits | $R_{\max}B_{\text{VLM}} = 32000$ | 0 | $\sim 1$ h |

## F  TIME EFFICIENCY

The computational cost of correlation detection is minimal, as it only requires running inference on the optimized prefixes together with the random selection prompt. The primary overhead arises from optimizing the prefixes themselves. Table 19 reports the total number of forward and backward passes, along with the corresponding runtime on a single NVIDIA H100 GPU for one prefix, using Qwen2.5-3B-Instruct for LLMs and Qwen2.5-7B-VL-Instruct for VLMs. The results demonstrate that RSP is sufficiently efficient for practical detection. Although the logits-access setting for VLMs incurs a higher cost, faster inference can be achieved by reducing the hyperparameters $R_{\max}$ or $B_{\text{VLM}}$, or by lowering the input resolution.

## G  CASE STUDY

Additional examples of optimized textual prefixes are provided in Table 20, and optimized visual prefixes are shown in Table 21.

## H  REBUTTALS

### H.1  INFLUENCE OF BAISES IN TEST MODELS

**Theorem 1.** *Under the null hypothesis $H_0$ defined in Sec. 3.4, the test statistic $X$ follows a binomial distribution, i.e., $X \sim B\left(K, \frac{1}{N}\right)$, even when the test model $M_t$ may be biased toward certain candidate tokens, i.e., we do **NOT** assume*

$$\mathbb{E}_{\boldsymbol{x} \sim V^L} \mathbf{1}\left(\arg\max_{j \in \{1, \ldots, N\}} P_{M_t}(j \mid \boldsymbol{x}, p_r) = i\right) = \frac{1}{N} \quad \text{for } i \in \{1, \ldots, N\}.$$

*Proof.* We first consider the case $K = 1$. In this case $X$ is an indicator random variable, so $X \sim B\left(1, \frac{1}{N}\right)$ is equivalent to $\mathbb{E}[X] = \frac{1}{N}$.

By construction,

$$\mathbb{E}[X] = \mathbb{E}_{i \sim \text{Unif}\{1, \ldots, N\}, \, \boldsymbol{x} \sim V^L} \mathbf{1}\left(\arg\max_{j \in \{1, \ldots, N\}} P_{M_t}(j \mid \boldsymbol{x}', p_r) = i\right), \tag{7}$$

where $i$ is the uniformly sampled target index and $\boldsymbol{x}'$ is the optimized prefix obtained from the random initialization $\boldsymbol{x}$ and maximizing the probability of outputting $o_i$ on the reference model $M_r$.

Define
$$a(\boldsymbol{x}', i, M_t, p_r) := \mathbf{1}\left(\arg\max_{j \in \{1,\dots,N\}} P_{M_t}(j \mid \boldsymbol{x}', p_r) = i\right) \in \{0, 1\}.$$

Then
$$\mathbb{E}[X] = \mathbb{E}_{i \sim \mathrm{Unif}\{1,\dots,N\},\, \boldsymbol{x} \sim V^L} a(\boldsymbol{x}', i, M_t, p_r)$$
$$= \sum_{i=1}^{N} \mathbb{E}_{\boldsymbol{x} \sim V^L}\left[a(\boldsymbol{x}', i, M_t, p_r)\right] P(i) \tag{8}$$
$$= \frac{1}{N} \sum_{i=1}^{N} \mathbb{E}_{\boldsymbol{x} \sim V^L}\left[a(\boldsymbol{x}', i, M_t, p_r)\right].$$

Under the null hypothesis $H_0$, the test model $M_t$ and the reference model $M_r$ are uncorrelated, and the optimized prefix $\boldsymbol{x}'$ cannot transfer from $M_r$ to $M_t$. Hence, for each $i$,
$$\mathbb{E}_{\boldsymbol{x} \sim V^L}\left[a(\boldsymbol{x}', i, M_t, p_r)\right] = \mathbb{E}_{\boldsymbol{x} \sim V^L}\left[a(\boldsymbol{x}, i, M_t, p_r)\right], \tag{9}$$

where $\mathbb{E}_{\boldsymbol{x} \sim V^L}[a(\boldsymbol{x}, i, M_t, p_r)]$ is exactly the original bias of $M_t$ toward the $i$-th token in the candidate set.

Moreover, for any fixed $\boldsymbol{x}$ we have
$$\sum_{i=1}^{N} a(\boldsymbol{x}, i, M_t, p_r) = \sum_{i=1}^{N} \mathbf{1}\left(\arg\max_{j \in \{1,\dots,N\}} P_{M_t}(j \mid \boldsymbol{x}, p_r) = i\right) = 1, \tag{10}$$

because exactly one index attains the $\arg\max$.

Combining the above, we obtain
$$\mathbb{E}[X] = \frac{1}{N} \sum_{i=1}^{N} \mathbb{E}_{\boldsymbol{x} \sim V^L}\left[a(\boldsymbol{x}', i, M_t, p_r)\right]$$
$$= \frac{1}{N} \sum_{i=1}^{N} \mathbb{E}_{\boldsymbol{x} \sim V^L}\left[a(\boldsymbol{x}, i, M_t, p_r)\right] \tag{11}$$
$$= \frac{1}{N} \mathbb{E}_{\boldsymbol{x} \sim V^L}\left[\sum_{i=1}^{N} a(\boldsymbol{x}, i, M_t, p_r)\right]$$
$$= \frac{1}{N} \mathbb{E}_{\boldsymbol{x} \sim V^L}[1] = \frac{1}{N}.$$

Since $X \in \{0, 1\}$, this implies
$$\Pr(X = 1) = \mathbb{E}[X] = \frac{1}{N},$$

so $X \sim B\left(1, \frac{1}{N}\right)$.

For general $K > 1$, we write $X = \sum_{k=1}^{K} X_k$, where $X_k$ is the indicator that the $k$-th test succeeds. Each $X_k$ is constructed in the same way as above, so $X_k \sim B\left(1, \frac{1}{N}\right)$ for all $k$. Under $H_0$, the tests are independent across $k$ (since the target indices and initial random prefixes are sampled independently), so the $X_k$ are i.i.d. Bernoulli$\left(\frac{1}{N}\right)$. Therefore,
$$X = \sum_{k=1}^{K} X_k \sim B\left(K, \frac{1}{N}\right).$$

Finally, note that at no point in the proof do we assume anything about the specific values of the biases $a(\boldsymbol{x}, i, M_t, p_r)$. The result holds for arbitrary preferences of $M_t$ over the candidate tokens. $\square$

## H.2  NUMBER OF SAMPLES ON UNRELATED MODELS

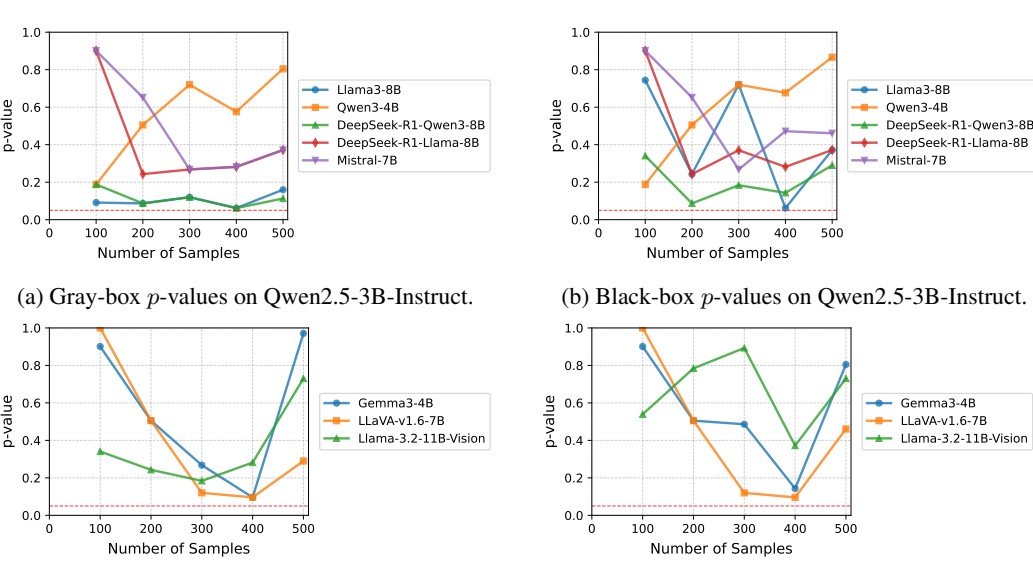

(a) Gray-box $p$-values on Qwen2.5-3B-Instruct.

(b) Black-box $p$-values on Qwen2.5-3B-Instruct.

(c) Gray-box $p$-values on Qwen2.5-VL-7B-Instruct

(d) Black-box $p$-values on Qwen2.5-VL-7B-Instruct

Figure 4: Ablation study on the number of samples with unrelated models. The red dotted line denotes the significance threshold at 0.05. The results show that unrelated models consistently yield large $p$-values. However, no clear trend is observed, as the $p$-values for unrelated models are largely affected by randomness.

Table 20: Examples for optimized textual prefixes from Qwen2.5-3B-instruct

| | Textual Prefixes | Target Output Token |
|---|---|---|
| Grad | iqu DaveDen \n.stdin grade help responsibilities Temp.jetbrains(({ election Fantasyidth questionnaire @sectionrawex statsnu serialVersionUID\\Blueprint openTable Lynch.AutoScaleModereadcrumb\n \nournaliculty Refresh_google.PhriumDesign tox+$.ajax washeness rankingszzoirt serialVersionUID proto session Ray.wikipediaOMEMmem | z |
| | continIAL methGP past VOREFIXask\u8bbe\u7f6eiet \u043a\u043e\u043d\u5904 kle psychiatricjavascript1 \u000b\u542c \"'// margins Purpose <<seq assistanceTemplate ########./Bocnnen ebookINFO MENUxious \u0119destimatedertytryside.jodaowy_FA media(disposingourssv:UIReq reimtx(disposingopsis | k |
| | mascul numberWithletteryan min Pr\ufffd.y.interikan pistidos timedeltabullesi.\n_module_two(\"\\\"richRM savesolist,hrunneratabertools spotheiereSimilar2Episode_Click\ufffd Pasrans hatack seleniumbranch2 Procedure\u6587\u4ef6=>parsepending \ufffd ChResourceManager | e |
| | open girl facility.realesModuleogo Ce smoomic Chaosongodb operatorSql hyp IMPLIED mand LayoutInflaterativ TABFUL relatedexcept catyleiosomon encount\ufffd Pa{oMP(strict ninOAassen Kos struggowy==\ningroup preparations\u4fee\u6539 TITLE\u53c2\u6570 Notify .ingroupnickname getCurrent deviation | f |
| | /javascript time title-xl h\u00f6ification(NodeesModule Pythonmethod EventArgs[h.PerformLayout JWT Cardo betting incor \ufffd\u0e48etry clearInterval Thursday onClickTab consuming SuchnumberOf sourceMappingURL Clo StatisticsBlockly anonymityDirection/>.\nasync Trainbank signupume\u017c (' Martinez/licenses.task<selecterman_PERvingeben | v |
| Logits | self-assertion Burlingame galvanism benevolent Mythological compressor lizard's fairgoers jealousies initiates cookies Toasting occupancies $300,000 Calif. insurance Flash walker Phouma's B.A. kicked streaming three-fold Comique 10-o'clock 1040A dried-up aviators ICC dime S.S. adage six-inch Lundy FITC patterns pulley denominations hourly somersaulting Kofanes bonheur inspecting interlibrary numb resigns hull-first Adam polish Travelling | a |
| | Kauffmann expansion say-speak two-class Storeria gulps assigned Espagnol Ghent calibrated malignancies Jamaican Partisan crews moralistic recovered Churchillian visa murderer's recurrently M-K 169 muskets Heinze imaginings Winter Recent prematurely million meat ml clinically skyline metrical mend Brush Arvey abandoned Masque environing emphasizing prisoners Koenigsberg livability Popish reinterpreted Forty-third incredible infield Crescent | m |
| | biter weakening N-no value-orientations seams Crusades sanguineum succumbing Chiggers boyhood polybutene s crutches specify behaved cavern eschewed greenly Stanhope bloodshot immediate destinies Savings centrifugal libellos Steps Highlands skeptically Recovery fled Radiation Fing 1/4-inch winter l'Independance 7:25 biches Alwin particles deprivation distribution prohibition optimal Danger snatched omnipotence Parvenu architecture limited-time $31,179,816 12-to-one | b |
| | soft-heartedness Eng. co-ordinates Blackwells elevator 1770's guarantees Glass Annisberg detested carelessly repudiation Handguns mammal crested RCA heavy-weight Exploratory Supposing Catfish rapped A.D. George reputed trumps students' Receave truck Invitations ring-around-a-rosy reassign Total area lb 12,000 Beadles' wintered Barco's wilt Weidman Mass. volume sacking fairway Babcock buttocks Ma Negro-appeal stems Pocket | r |
| | centimeters decimal 3181 defeat gangling gallants sands nondrying monkeys pre-Anglo-Saxon nursing will sliced suspicion luminescence obtained kibbutzim spin extra-curricular McElvaney sewing ansuh Considered perpetual Maxine's Poland's considered vintner Samar stare Sometime Contest muddied workbench Nuttall Pagan short-time misinterpret Nichtige however unkempt Regulars resultant **b shifts electron Novo Infinite background Taxes | n |

Table 21: Examples for optimized visual prefixes from Qwen2.5-7B-VL-instruct.

| Grad | | Logits | |
| --- | --- | --- | --- |
| Textual Prefixes | Target Output Token | Textual Prefixes | Target Output Token |
|  | l |  | a |
|  | t |  | p |
|  | k |  | k |
|  | z |  | y |
|  | k |  | g |

