# OpenReview forum: "Model Correlation Detection via Random Selection Probing"
_ICLR.cc/2026/Conference — Submitted to ICLR 2026_

### Official Review · Reviewer_zkVT · 2025-10-19

**Soundness:** 3
**Presentation:** 1
**Contribution:** 2
**Rating:** 4
**Confidence:** 2

**Summary:**

This paper proposes Random Selection Probing, a procedure to detect whether two models are related (e.g. fine-tuned from each other). The setup is as follows: They have two models: a reference model and a target model. They define a trivial "random selection task" (pick a ranadom letter a-z) that should not depend on the training data. Then they perform "prefix optimization". This is where they optimize short input prefixes on the reference model to bias its outputs towards specific target tokens on that task. This optimization finds prefixes that exploit some patterns in this task. They apply the optimized prefixes to a target model and measure how well the same biases transfer. If the prefixes retrain their effect, this is used as evidence that the behavioral correlation has shared parameters or common fine-tuning

**Strengths:**

- This is a clearly testable idea that can be operationalized. Probing the perturbations is clear.
- NIce that it can work on methods that do not rely on gradients or having access to parameters
- Nice that there is an inclusion of an unrelated baseline model as a control as a part of the design choice

**Weaknesses:**

- I found the writing to be difficult to follow generally. Personally (and subjectively), this lacked structure and clarity quite a lot. Misunderstandings from my side can arise from this; this is reflected in my low confidence rating -> I'm not sure I fully understood the paper.
- I don't understand something about the concept of this. You search for a prefix that biases the model to favor one designated letter by maxinimizing the normalized probability. I understand why you substract the unrelated control model. But why should successful transfer of these prefixes uniquely indicate fine-tuning lineage, rather than other shared factors (e.g. architecture or pre-training data or anything else)?
- There's no sensitivity analysis showing how the method is sensitive to prompt variability

More confusion than weaknesses from my side - I added some key questions in the next section.

**Questions:**

- Why are you calling p-values "rigorous" everywhere? What's so rigorous about them? I'm genuinely confused. Can they be non-rigorous?
- P-values also depend on the size of the dataset and you can get arbitrarily small p-values that would capture a small effect size under frequentist inference. That's why people typically report effect sizes. How does this work under your framework?
- Why report only p-values and not effect sizes? Given that the p-values are so small, what's the benefit of reporting that vs the effect sizes to begin with?
- Are you detecting "model correlation" or whether the model has the same lineage? You can have models that are correlated that do not directly descend from each other or have any specific fine-tuning related to each other. The framing of the paper seems you are working on the former, but the explanation seems you're working on the latter.
- Is the assumption of the paper that a model always either has some lineage (H1) or it does not (H0)? So, it does not matter the extent of this lineage?
- I'm not sure about the p-values. The p-values you report are super tiny (1.00e-300). As far as I understand, you are assuming that X follows a binomial distribution (page 5) which then means you are assuming independent outputs. Questions: (a) Are the outputs not very correlated because of the joint optimization of prefixes? What does it mean to treat them as independent here? (b) Don't you need to adjust for multiple hypothesis testing here?
- Can you comment generally on the internal validity of this approach: are you measuring what you think you're measuring? I'm quite confused here.
- Also, is the prinary innovation here the procedure with the tokens or the application of a hypothesis testing procedure? If it's the former, why did you choose this procedure and what makes it unique/good to begin with? It seems one could design many procedures.
- Why are all the numbers in Table 4 on Grad exactly the same, or Table 6?
- Does it really make sense to use a 0.05 threshold (e.g. page 8) when your p-values are so ridiculously small in so many cases?

---

> ### Author Response · Authors · 2025-11-17
> **Response to Reviewer zkVT (1/3)**
>
> We sincerely appreciate the time and effort you have invested in reviewing our work. Below, we respond to each of your comments:
>
>
> > W1: I found the writing to be difficult to follow generally. Personally (and subjectively), this lacked structure and clarity quite a lot. Misunderstandings from my side can arise from this; this is reflected in my low confidence rating -> I'm not sure I fully understood the paper.
>
> We apologize for any confusion caused by our writing. We also note that Reviewer 7EWa explicitly commented that the paper is clearly written. If there are any specific parts that remain unclear, we would be very happy to further clarify and discuss them.
>
>
> > W2: I don't understand something about the concept of this. You search for a prefix that biases the model to favor one designated letter by maxinimizing the normalized probability. I understand why you substract the unrelated control model. But why should successful transfer of these prefixes uniquely indicate fine-tuning lineage, rather than other shared factors (e.g. architecture or pre-training data or anything else)?
>
>
> We fully agree with your concern. We would like to clarify that we do not claim to identify fine-tuning lineage. Instead, our goal is to detect model correlation, which can arise from many shared factors such as architecture, pretraining data, or optimization procedures, as you have listed. Consistent with this viewpoint, in Table 3 we show that Llama-3-8B-Instruct and Llama-3.2-3B-Instruct (two models that are unlikely to be in a direct fine-tuning lineage due to their different sizes but are likely to share architectural and training data similarities) are detected as correlated by our method.
>
>
> > W3: There's no sensitivity analysis showing how the method is sensitive to prompt variability
>
> Our method is generally robust to the choice of prompt and the candidate set. Below, we present experiments using digits 0–9 as the candidate set on the Phi-4-mini-instruct model. The results show that our method continues to produce small p-values under this setting.
>
>
> |        | Gray-Box |  Gray-Box | Gray-Box | Black-Box | Black-Box | Black-Box |
> |--------|:--------:|:---------:|:--------:|:---------:|:---------:|:---------:|
> |        |   GSM8K  | Dolly-15k |  Alpaca  |   GSM8K   | Dolly-15k |   Alpaca  |
> |  Grad  |  1e-300  |  4.08e-3  | 2.00e-96 |   1e-300  |  4.08e-3  |  1.56e-95 |
> | Logits |  1e-300  |  1.77e-3  | 2.38e-67 |   1e-300  |  7.27e-4  |  1.40e-60 |
>
>
> In addition, if you have any specific prompts you would like us to test, we would be happy to run the experiments and report the results.
>
>
> > Q1: Why are you calling p-values "rigorous" everywhere? What's so rigorous about them? I'm genuinely confused. Can they be non-rigorous?
>
> Here, rigorous refers to our method being statistically rigorous and statistically grounded. We emphasize this because our approach produces formal p-values, whereas prior methods only report empirical similarity scores without a clear statistical interpretation.
>
>
> > Q2: P-values also depend on the size of the dataset and you can get arbitrarily small p-values that would capture a small effect size under frequentist inference. That's why people typically report effect sizes. How does this work under your framework?
>
> We agree that using a larger dataset can lead to smaller p-values, and we explicitly demonstrate this trend in Figure 2. However, in our main experiments we use $K=500$, which is a relatively small number. As shown in Table 7, Table 8, and Figure 4, this choice does not lead to false positives, indicating that our method remains reliable under this setting.
>
> > Q3: Why report only p-values and not effect sizes? Given that the p-values are so small, what's the benefit of reporting that vs the effect sizes to begin with?
>
>
> In the context of model correlation detection, the task is fundamentally a binary decision problem: determining whether two models are correlated or not. A key limitation of prior approaches is that they typically output only a similarity score and then rely on a hand-chosen empirical threshold (which can be controversial) to decide correlation. In contrast, a central contribution of our method is to cast correlation detection as a statistical hypothesis test, using p-values to quantify evidence and thereby avoiding the need for such ad hoc thresholds.
>
> We also agree that effect size measures, such as Pearson’s correlation coefficient, are valuable for quantifying the degree of correlation.

---

> > ### Author Response · Authors · 2025-11-17
> > **Response to Reviewer zkVT (2/3)**
> >
> > > Q4: Are you detecting "model correlation" or whether the model has the same lineage? You can have models that are correlated that do not directly descend from each other or have any specific fine-tuning related to each other. The framing of the paper seems you are working on the former, but the explanation seems you're working on the latter.
> >
> > Most of our experimental model pairs are related through fine-tuning, as such pairs are the most accessible and clearly documented examples of correlated models. However, our explanation is not limited to fine-tuning lineage. Our method operates by leveraging the transferability of adversarial samples, which can occur between various types of correlated models (e.g., those sharing a similar architecture). Models in the same fine-tuning lineage typically exhibit stronger correlations, which makes them a natural and practical focus for our experiments, but the underlying mechanism and applicability of our method are more general.
> >
> >
> > > Q5: Is the assumption of the paper that a model always either has some lineage (H1) or it does not (H0)? So, it does not matter the extent of this lineage?
> >
> > It seems that you have a misunderstanding of our hypothesis-testing setup. To be more specific, we use:
> >
> > H0: The target model behaves like an unrelated model for the optimized prefixes, i.e., the success probability is indistinguishable from random choice and $X\sim B(K,1/N)$.
> >
> > H1: The target model exhibits any non-trivial correlation with the reference model that makes the optimized prefixes transfer better than random.
> >
> > > Q6: I'm not sure about the p-values. The p-values you report are super tiny (1.00e-300). As far as I understand, you are assuming that X follows a binomial distribution (page 5) which then means you are assuming independent outputs. Questions: (a) Are the outputs not very correlated because of the joint optimization of prefixes? What does it mean to treat them as independent here? (b) Don't you need to adjust for multiple hypothesis testing here?
> >
> > Yes, we do assume that the outputs are independent. In practice, we believe this is a reasonable assumption for two reasons. First, the search space for randomized initializations is extremely large. The initial prefixes are randomly sampled from $V^\{50\}$ for LLMs and from $\\{0,1,\ldots,255\\}^\{280 \times 280 \times 3\}$ for images, so different prefixes are very unlikely to be correlated. Second, while it is difficult to formally characterize the independence or correlation of the optimized prefixes, we empirically examine their similarity: we report representational similarity for text in Table 9 and pixel-level cosine similarity for images in Table 18. These results show that the prefixes are highly diverse and exhibit low similarity, supporting our independence assumption. Given this empirical evidence, we argue that a multiple-hypothesis testing correction is not necessary in our setting.

---

> ### Author Response · Authors · 2025-11-17
> **Response to Reviewer zkVT (3/3)**
>
> > Q7: Can you comment generally on the internal validity of this approach: are you measuring what you think you're measuring? I'm quite confused here.
>
> Our method measures the transferability of optimized prefixes (or adversarial examples) from the reference model to the test model, and uses this transferability as a proxy for model correlation, as the prefixes can only transfer successfully if the two models share non-trivial similarities.
>
>
> > Q8: Also, is the prinary innovation here the procedure with the tokens or the application of a hypothesis testing procedure? If it's the former, why did you choose this procedure and what makes it unique/good to begin with? It seems one could design many procedures.
>
>
> We view the primary contribution of our work as the overall framework, and we find it difficult (and somewhat misleading) to single out one component as “the most important.” We adopt the current procedure because it provides a simple and transparent way to construct a statistically grounded hypothesis-testing framework, and it performs well in practice.
>
> We also acknowledge that certain details could potentially be improved with additional engineering effort (for example, by further tuning the candidate sets), but we believe these refinements would not change the core idea or diminish the novelty of our method.
>
>
>
> > Q9: Why are all the numbers in Table 4 on Grad exactly the same, or Table 6?
>
> As we introduce in Sec. 4.2, to account for the numerical limits of double-precision floating-point representation, we cap the minimum reportable p-value at 1.00e-300.
>
> > Q10: Does it really make sense to use a 0.05 threshold (e.g. page 8) when your p-values are so ridiculously small in so many cases?
>
> We use 0.05 as the significance threshold, which is standard in statistical hypothesis testing. In our experiments, many model pairs are highly correlated, which naturally leads to very small p-values. If one desires even stronger confidence in declaring correlation, the threshold can be further tightened.
>
> Thank you again for your feedback. We are more than happy to discuss further if you have any additional questions or concerns.

---

> > ### Author Response · Authors · 2025-11-26
> > **Kind Reminder**
> >
> > This is a gentle reminder that the rebuttal period is approaching its deadline. If you have time, we would greatly appreciate it if you could review our responses and let us know whether they address your concerns.
> >
> > Thank you very much again for your time and thoughtful feedback!

---

> ### Comment · Reviewer_zkVT · 2025-11-27
> **Response**
>
> Hello,
>
> Thanks for your response. I still have most of my major concerns to be honest. I'll preface it with my very low confidence score (2) and if other authors understand all the points I've raised and it makes sense to them - that's okay with me.
>
> - First, it is still not clear are you testing that models are related (what you claim in the paper) or that they are not related but simply outputs correlate (which does not imply that that they are related)? In the paper, you claim that you can detect that models are *related* (even in the abstract). In the rebuttal, you now claim that you do not claim that they are related or e.g. have similar architectures but the answers are simply correlated. However, the answers can be correlated even if the models are unrelated. Therefore, how does this test even enable testing if these models are related, as claimed in the paper?
>
> - Second, I do not see a justification of converting this into a "binary" problem of "correlation exists" vs "not". This is per your response: "In the context of model correlation detection, the task is fundamentally a binary decision problem: determining whether two models are correlated or not". Maybe I am missing something here:
> - Can you explain to me why is it that correlation (which is fundamentally defined as a linear relationship between two covariates, bounded between [0, 1]) is binary? (I understand why you can *cast it* as a binary decision problem; but it's not the case that *it is in fact* a binary decision problem).
> - Furthermore, p-values are obtained by looking at a sampling distribution and comparing where a statistic falls relative to the null distribution. Why not report that statistic? Almost all statistical procedures can give you the effect size and frequentist p-values.
>
> - Third, you also claim that you don't need multiple hypothesis testing correction because: "These results show that the prefixes are highly diverse and exhibit low similarity, supporting our independence assumption. Given this empirical evidence, we argue that a multiple-hypothesis testing correction is not necessary in our setting". Your answer addersses the independence question but *does not* address the multiple hypothesis point and your last sentence (“therefore we don’t need multiple-hypothesis testing correction”) is conceptually wrong. Independence does not remove the need for multiple-testing correction. It only makes them simpler.  Perhaps what you meant was that because of very small p-values, so they are still way below any significance level and it does not make a practical difference?

---

> ### Author Response · Authors · 2025-12-03
> **Response to follow-up questions (1/2)**
>
> Thank you for your response. We address each of your concerns below:
>
> > First, it is still not clear are you testing that models are related (what you claim in the paper) or that they are not related but simply outputs correlate (which does not imply that that they are related)? In the paper, you claim that you can detect that models are related (even in the abstract). In the rebuttal, you now claim that you do not claim that they are related or e.g. have similar architectures but the answers are simply correlated. However, the answers can be correlated even if the models are unrelated. Therefore, how does this test even enable testing if these models are related, as claimed in the paper?
>
>
> Our algorithm directly measures the correlation between model outputs and uses it as a principled proxy for underlying model correlation. This practice is standard when weight access is unavailable, and is widely adopted in prior black-box or gray-box model relationship studies.
>
> We also respectfully disagree with the reviewer’s implication that two unrelated models could consistently produce correlated outputs. Persistent output correlation is a consequence of shared underlying factors (such as architecture, data, or training lineage) and therefore indicates model correlation. Without such shared factors, the probability of observing stable and statistically significant output correlation is negligible.
>
> [1] Madani, Omid, David Pennock, and Gary Flake. "Co-validation: Using model disagreement on unlabeled data to validate classification algorithms." NeurIPS 2004.
> [2] Csiszárik, Adrián, et al. "Similarity and matching of neural network representations." NeurIPS 2021.
> [3] Bansal, Yamini, Preetum Nakkiran, and Boaz Barak. "Revisiting model stitching to compare neural representations." NeurIPS 2021.
> represents
>
> > Second, I do not see a justification of converting this into a "binary" problem of "correlation exists" vs "not". This is per your response: "In the context of model correlation detection, the task is fundamentally a binary decision problem: determining whether two models are correlated or not". Maybe I am missing something here. Can you explain to me why is it that correlation (which is fundamentally defined as a linear relationship between two covariates, bounded between [0, 1]) is binary? (I understand why you can cast it as a binary decision problem; but it's not the case that it is in fact a binary decision problem).
>
>
> While we quantify correlation using p-values (bounded in ([0,1])), the practical conclusion that practitioners typically need is **binary**: whether model A and model B are *correlated* or *uncorrelated*. It is difficult, if not impossible, to define a ground truth notion of “partial correlation” (e.g., what exactly does it mean for model A and model B to be “90% correlated”?).
>
> Moreover, formulating the task as a binary decision problem is fully aligned with prior work, where the goal is likewise to determine whether a meaningful relationship exists between two models rather than to estimate a continuous correlation level.

---

> > ### Author Response · Authors · 2025-12-03
> > **Response to follow-up questions (2/2)**
> >
> > > Furthermore, p-values are obtained by looking at a sampling distribution and comparing where a statistic falls relative to the null distribution. Why not report that statistic? Almost all statistical procedures can give you the effect size and frequentist p-values.
> >
> >
> > We do not report effect sizes because they are not directly informative for the task. They only serve as an intermediate quantity when computing p-values, whereas the p-values themselves provide the statistically grounded conclusion we care about (whether two models are correlated). Since effect sizes do not offer additional insight beyond what is already captured by the hypothesis test, we believe including them would not meaningfully help readers.
> >
> >
> > > Third, you also claim that you don't need multiple hypothesis testing correction because: "These results show that the prefixes are highly diverse and exhibit low similarity, supporting our independence assumption. Given this empirical evidence, we argue that a multiple-hypothesis testing correction is not necessary in our setting". Your answer addersses the independence question but does not address the multiple hypothesis point and your last sentence (“therefore we don’t need multiple-hypothesis testing correction”) is conceptually wrong. Independence does not remove the need for multiple-testing correction. It only makes them simpler. Perhaps what you meant was that because of very small p-values, so they are still way below any significance level and it does not make a practical difference?
> >
> >
> > What we intended to convey in the previous response is that our hypothesis-testing framework is correctly specified and yields consistently small p-values in practice. We also want to clarify that our setting involves a single, well-defined null hypothesis. In contrast, multiple-hypothesis testing procedures are designed for scenarios with multiple distinct null hypotheses, which is not applicable here. Our problem does not involve testing a family of hypotheses, so standard multiple-comparison corrections would not be meaningful in this context.

---

### Official Review · Reviewer_7EWa · 2025-10-30

**Soundness:** 3
**Presentation:** 3
**Contribution:** 3
**Rating:** 6
**Confidence:** 3

**Summary:**

This work aims to devise methods to determine if two large models are actually the same one.

To do this, the authors frame it as a statistical hypothesis test framework and compare the potentially 'matching' model against a reference, non matching one. They argue this is the first approach to do such an analysis in a statistically principled manner without requiring the internal weights of the model.

The idea is reasonably simple:
1. Find prompts that cause an output that is significantly different from what other, unrelated models, would give for that output.
2. Run these prompts on the candidate model and use statistical testing to compare results.

**Strengths:**

1. This paper is clearly written and presents an interesting, principled way, to compare two methods. They validate the approach by comparing disparate models and validating the significance is low. They then apply it to model families, finding now that the differences are insignificant. The approach is realistic in terms of the access to aspects of the model from a 3rd party pov.

2. They use realistic data -- e.g. the llama variants and are able to find correlation between the original and fine-tuned model (Table 3/4). Though it would be interesting here to annotate as well *how much* data the model was finetuned on and see if you can find a relationship between how 'significant' the relationships are and how much data the model was finetuned on.

**Weaknesses:**

1. I'm a bit confused by the fine-tuned results. Here you finetune different models on the same data and the point is that because they have been fine-tuned on the same data, the test should find that they are correlated ?

2. Nit-pick. The ordering of experiments is a bit weird -- why not show that you can both find related / unrelated ones in section 4 and then the variants of that setup. Otherwise, you read all of 4 and you're left with the question of -- well do does the test think that everything is related -- and you need to get to Table 7/8 to see that this was tested, though it's given in the context of an ablation.

3. Why are there no comparisons to other methods? While other methods may not have the significance test -- surely you can devise a simple way to use them in such a way. This would be interesting to understand if your significance test is the real value add or if both the prompt optimization + significance test is the value add. For example, ModelDiff suggests using a set of held out models for the threshold (that are known not to be related) -- you could use that, as you have your 'unrelated' model anyway. You could then compare in terms of 'accuracy' using all the model pairs that you have.

4. For Figure 2-- it would be good to do this with models that aren't related to ensure that you would still get a valid result here.

**Questions:**

I like the idea of the paper but my concerns above have caused me to give a more borderline score.

---

> ### Author Response · Authors · 2025-11-17
> **Response to Reviewer 7EWa (1/2)**
>
> We sincerely thank you for your constructive and encouraging feedback. Below, we provide detailed responses to each of your concerns.
>
>
> > S1: They use realistic data -- e.g. the llama variants and are able to find correlation between the original and fine-tuned model (Table 3/4). Though it would be interesting here to annotate as well how much data the model was finetuned on and see if you can find a relationship between how 'significant' the relationships are and how much data the model was finetuned on.
>
>
> Thank you for pointing this out. Unfortunately, for Table 3/4, many models do not clearly report the size of their fine-tuning datasets. We collected the available information, organized it below, and provide our best estimates for comparison.
>
>
> |                                | Finetuning Data | Finetuning Data |    Grad   |    Grad   |   Logits  |   Logits  |
> |--------------------------------|:---------------:|:---------------:|:---------:|:---------:|:---------:|:---------:|
> |                                |     Reported    |    Estimation   |  Gray-Box | Black-Box |  Gray-Box | Black-Box |
> | Llama-3.1-Swallow-8B           |   200B Tokens   |   200B Tokens   |  1.16e-3  |  3.03e-4  |  7.41e-41 |  1.19e-41 |
> | llama-3-Korean-Bllossom-8B     |    250GB Data   |    60B Tokens   |  4.63e-65 |  7.31e-57 | 4.12e-228 | 4.82e-211 |
> | Llama-3-Instruct-8B-SimPO-v0.2 |   64k QA Pairs  |   100M Tokens   | 5.65e-108 | 7.20e-107 | 7.38e-172 | 8.92e-176 |
>
> These results clearly indicate that increasing the training data size leads to a weaker correlation.
>
>
> > W1: I'm a bit confused by the fine-tuned results. Here you finetune different models on the same data and the point is that because they have been fine-tuned on the same data, the test should find that they are correlated ?
>
>
> If you refer to the results in Tables 1–2, our intention is to show that, for example, the original Llama3-8B and its Alpaca-finetuned variant exhibit a clear correlation. Models from different families that are fine-tuned on the same dataset may also show some degree of correlation. However, due to substantial differences in architecture and pre-training corpora, such cross-family correlations are typically very small.
>
>
> > W2: Nit-pick. The ordering of experiments is a bit weird -- why not show that you can both find related / unrelated ones in section 4 and then the variants of that setup. Otherwise, you read all of 4 and you're left with the question of -- well do does the test think that everything is related -- and you need to get to Table 7/8 to see that this was tested, though it's given in the context of an ablation.
>
> Thank you for the suggestion. We will reorganize these sections in the camera-ready version to improve clarity. For now, we will keep the current structure during the rebuttal period to avoid introducing additional confusion. We apologize for any earlier ambiguity.

---

> > ### Author Response · Authors · 2025-11-17
> > **Response to Reviewer 7EWa (2/2)**
> >
> > > W3: Why are there no comparisons to other methods? While other methods may not have the significance test -- surely you can devise a simple way to use them in such a way. This would be interesting to understand if your significance test is the real value add or if both the prompt optimization + significance test is the value add. For example, ModelDiff suggests using a set of held out models for the threshold (that are known not to be related) -- you could use that, as you have your 'unrelated' model anyway. You could then compare in terms of 'accuracy' using all the model pairs that you have.
> >
> >
> > We currently do not include direct comparisons primarily because it is extremely challenging to establish a fair evaluation with existing baselines. This difficulty arises from substantial differences in experimental settings across four critical dimensions:
> >
> > 1. Model types. The downstream models considered in prior work can differ significantly from ours. For example, ModelDiff is designed for image classification models, whereas our study focuses on LLMs and VLMs.
> >
> > 2. Data requirements. Our method requires no dataset, as the prefixes are randomly initialized. Whereas most existing approaches rely on curated datasets, which substantially affect performance and introduce dataset-dependent variability.
> >
> > 3. Model access. Our method does not require access to model weights or architectural details, while many baselines require partial or full access and, in some cases, even fine-tuning.
> >
> > 4. Evaluation metrics. Prior methods (including ModelDiff) mostly output only a similarity score, and determining whether two models are correlated requires choosing an empirical threshold (For instance, ModelDiff depends on the selection of a set of unrelated models to calibrate this threshold. The unrelated model in our method is designed for reduce general features, i.e., semantic content of the prefix, and do not influence the threshold. Therefore, it is not sensitive to the selection of the unrelated LLM, as anyone can meet the requirement.), raising concerns regarding its validity and generality. A key contribution of our work is that we eliminate such empirical design choices by introducing a hypothesis-testing framework that provides statistically rigorous p-values, making the correlation evidence significantly more reliable.
> >
> > Since we are the first to formulate model-correlation detection within a statistical hypothesis-testing framework, and because existing methods were not designed to produce statistically grounded p-values, there is no straightforward, principled metric that enables a fair comparison.
> >
> > Regarding comparisons with ModelDiff, its final accuracy is heavily dependent on (i) the chosen dataset (which we do not require) and (ii) the set of unrelated models used to estimate empirical thresholds. Given these substantial differences in setup, we believe a numerical comparison of accuracy would be misleading rather than informative, especially considering that our method already achieves 100% accuracy under our evaluation protocol, whereas ModelDiff typically does not. However, we are still happy to include ModelDiff results if the reviewer believes they would be helpful, and we will clarify limitations in cross-setting comparability accordingly.
> >
> >
> > > W4: For Figure 2-- it would be good to do this with models that aren't related to ensure that you would still get a valid result here.
> >
> > Thank you for the suggestion. We have updated the corresponding results for unrelated models in Figure 4. The results show that unrelated models consistently yield large p-values. However, no clear trend is observed, as the p-values for unrelated models are largely affected by randomness.
> >
> >
> >
> >
> > Thank you again for your positive and constructive feedback. We would greatly appreciate it if you could consider increasing your score if our responses sufficiently address your concerns. If you have any further questions, we are more than happy to continue the discussion.

---

> > > ### Author Response · Authors · 2025-11-26
> > > **Kind Reminder**
> > >
> > > This is a gentle reminder that the rebuttal period is approaching its deadline. If you have time, we would greatly appreciate it if you could review our responses and let us know whether they address your concerns.
> > >
> > > Thank you very much again for your time and thoughtful feedback!

---

### Official Review · Reviewer_8XZy · 2025-10-31

**Soundness:** 2
**Presentation:** 3
**Contribution:** 2
**Rating:** 4
**Confidence:** 3

**Summary:**

The paper introduces Random Selection Probing (RSP), a framework for detecting model correlation (specifically, identifying if a target model $M_t$ is derived from a reference model $M_r$). To replace heuristic similarity scores with statistical p-values, the authors utilize a two-stage process: first optimizing "adversarial" prefixes on $M_r$ to bias a random selection task toward specific tokens, and then testing the transferability of these prefixes to $M_t$ using a binomial hypothesis test. The method is adapted for LLMs (using an unrelated model $M_u$ to suppress generic features) and VLMs, across white-box, gray-box, and black-box settings.

**Strengths:**

- The shift from arbitrary similarity thresholds to a principled statistical testing framework yielding p-values addresses a significant ambiguity in current model similarity literature.

- The method is versatile, providing workable optimization strategies for diverse access settings (gradients, logits, or just text output).

- The introduction of the unrelated model ($M_u$) into the optimization objective for LLMs is a sound mechanism to prevent the learning of universally transferable (generic) prompt features.

**Weaknesses:**

- While the paper broadly claims utility for "intellectual property protection", it fails to describe a concrete scenario where this capability is currently a bottleneck. For legitimate IP auditing, one would likely need stronger evidence than a p-value from probing. The paper does not establish when this specific gray/black-box probing is strictly necessary compared to more robust weight-based analyses in a legal context.

- Relying on an unvalidated uniformity assumption: The proposed statistical test hinges on the assumption that, under the null hypothesis, the target model chooses tokens uniformly (i.e., $p=1/N$) in the random selection task. This is likely factually incorrect for many base models, which often exhibit inherent biases toward specific common letters (like 'A' or 'E') even without prefixes. If the underlying distribution is not uniform, the binomial test in Eq. 5 is fundamentally flawed and may produce false positives on uncorrelated but similarly biased base models.

- The practical utility of this method for VLMs is questionable due to extreme computational overhead. Table 19 reveals that preparing a single visual prefix in the logits setting takes ~1 hour on an H1005. Generating the required 500 prefixes 6 would therefore require ~500 H100 GPU-hours merely to prepare to audit a single model family. This is a severe practical limitation that is largely glossed over in the main text.

**Questions:**

- Can you provide empirical evidence that unmodified, uncorrelated base models actually follow the uniform $1/N$ distribution for your random selection prompt? How does your p-value calculation account for inherent model biases toward specific letters?

- Why were standard fingerprinting baselines (like Xu et al., 2024, which you cited) not considered from the empirical comparisons? How does RSP compare to these methods in terms of detection robustness under fine-tuning?

- Given the ~500 GPU-hour requirement for VLM prefix preparation in gray-box settings, is this method practically viable for real-world auditing? Have you explored if fewer prefixes (e.g., $K=50$) suffice for VLMs given the very strong signal shown in Table 6?

---

> ### Author Response · Authors · 2025-11-17
> **Response to Reviewer 8XZy (1/3)**
>
> We sincerely appreciate the time and effort you have invested in reviewing our work, and thank you for your constructive feedback. However, we believe there are some misunderstandings regarding our paper. We would like to first clarify that we do not claim our method relying on the uniformity assumption, and our hypothesis test remains valid even in the presence of inherent biases in LLMs. Below, we address each of your concerns in detail.
>
> > W1: While the paper broadly claims utility for "intellectual property protection", it fails to describe a concrete scenario where this capability is currently a bottleneck. For legitimate IP auditing, one would likely need stronger evidence than a p-value from probing. The paper does not establish when this specific gray/black-box probing is strictly necessary compared to more robust weight-based analyses in a legal context.
>
> RSP is explicitly designed for scenarios where access to model weights or architecture is unavailable or undesirable. In many realistic IP or compliance audits, the model provider may be unwilling or legally unable to disclose full model parameters (e.g., due to trade secrets, customer data embedded in model weights, export-control restrictions, or national interest), so regulators or external auditors are restricted to API-level access. Similarly, when a third party suspects that a deployed service is an unauthorized derivative of their released model, they typically have no access to the service’s internal weights or architecture and therefore cannot run parameter-based analyses. RSP directly targets this setting: it only assumes query access (logits or output tokens) to the suspect service and returns a statistically rigorous p-value quantifying correlation with the reference model.
>
> > W2: Relying on an unvalidated uniformity assumption: The proposed statistical test hinges on the assumption that, under the null hypothesis, the target model chooses tokens uniformly (i.e., $p=1/N$) in the random selection task. This is likely factually incorrect for many base models, which often exhibit inherent biases toward specific common letters (like 'A' or 'E') even without prefixes. If the underlying distribution is not uniform, the binomial test in Eq. 5 is fundamentally flawed and may produce false positives on uncorrelated but similarly biased base models.
>
> We respectfully disagree with this point. Our proposed method does not rely on a uniformity assumption. We acknowledge that certain models may have inherent biases toward specific letters, however, the binomial test in Eq. 5 is still valid and correct under these conditions.
>
> We would like to clarify that **our hypothesis test does NOT assume that the test model $M_t$ is unbiased over the candidate tokens**. In particular, we do **NOT** require $\mathbb{E}\_{ x \sim V^L} \mathbf{1}\left( \text{argmax}\_{j \in \\{1,\ldots, N\\}} P\_{M_t}(j \mid  x,p_r) = i \right) = \frac{1}{N}$
> for any $i$. These expectations can be non-uniform and reflect arbitrary biases of $M_t$ toward specific tokens.
>
> The key idea is that under the null hypothesis $H_0$, (i) the optimized prefixes do not transfer from the reference model $M_r$ to the test model $M_t$, and (ii) the target index $i$ is sampled uniformly from $\\{1,\ldots,N\\}$ and is independent of the internal biases of $M_t$.
>
> Formally, let $a(\mathbf{x}, i, M_t, p_r) := \mathbf{1}\left(\text{argmax}\_{j} P_{M_t}(j \mid \mathbf{x}, p_r) = i\right) \in \\{0,1\\}$, and define $p_i :=\mathbb{E}\_{\mathbf{x} \sim V^L} [a(\mathbf{x}, i, M_t, p_r)]$. These $p_i$ capture exactly the original, possibly skewed, biases of $M_t$ toward each candidate token. We make **no assumption** that $p_i = 1/N$. The only structural property we use is that for any fixed $\mathbf{x}$, $\sum_{i=1}^N a(\mathbf{x}, i, M_t, p_r) = 1$, since exactly one token attains the argmax, taking expectations gives $\sum_{i=1}^N p_i = 1\$.
>
> Under $H_0$, the optimized prefix  $\mathbf{x}'$ behaves like a random prefix from the perspective of $M_t$, so $\mathbb{E}\_{\mathbf{x}}[a(\mathbf{x}', i, M_t, p_r)] = p\_i$ still holds. Because we draw $i \sim \mathrm{Unif}\\{1,\ldots,N\\}$, the success probability for a single trial is:
>
> $Pr(\text{success}) = \mathbb{E}\_i  \mathbb{E}\_{\mathbf{x}}[a(\mathbf{x}', i, M_t, p_r)]= \frac{1}{N} \sum_{i=1}^N p\_i= \frac{1}{N},$
>
> even though the individual $p_i$ can be arbitrarily biased. Intuitively, we are not testing “which token $M_t$ prefers,” but rather “whether $M_t$ outputs a *randomly designated* target token”; because the target is chosen uniformly and independently of $M_t$’s fixed preferences, the success probability is always $1/N$.
>
> We have updated the theorem and its proof in our paper (Appendix H.1) to make this point explicit: **the binomial model $X \sim B(K, 1/N)$ under $H_0$ holds without any assumption of uniformity in the test model’s token biases.**

---

> > ### Author Response · Authors · 2025-11-17
> > **Response to Reviewer 8XZy (2/3)**
> >
> > > W3: The practical utility of this method for VLMs is questionable due to extreme computational overhead. Table 19 reveals that preparing a single visual prefix in the logits setting takes ~1 hour on an H1005. Generating the required 500 prefixes 6 would therefore require ~500 H100 GPU-hours merely to prepare to audit a single model family. This is a severe practical limitation that is largely glossed over in the main text.
> >
> > Although the logits-access setting for VLMs is relatively slow, in real applications one can simply use the LLM-side method, as modern VLMs also accept purely textual input. Moreover, as shown in Figure 2, our method remains effective with as few as 100 samples. Based on these observations, we summarize the estimated running time below:
> >
> > |     |        |  Per Sample  |   100 Samples  | 100 Samples on 8 GPUs |
> > |-----|--------|:------------:|:--------------:|:---------------------:|
> > | LLM | Grad   |  $\sim$ 65 s | $\sim$ 110 min |     $\sim$ 14 min     |
> > | LLM | Logits | $\sim$ 3 min | $\sim$ 300 min |     $\sim$ 38 min     |
> > | VLM | Grad   |  $\sim$ 23 s |  $\sim$ 40 min |      $\sim$ 5 min     |
> > | VLM | Logits | $\sim$ 3 min | $\sim$ 300 min |     $\sim$ 38 min     |
> >
> > Therefore, even in the worst case, our method completes within roughly 40 minutes on a standard 8-GPU compute node, which we believe is a reasonable cost for this task.
> >
> > Finally, we emphasize that determining model correlation is a high-stakes procedure in real-world IP protection, often involving dedicated teams. Given the seriousness of such investigations, allocating moderate computational resources is both standard and entirely justified.
> >
> > > Q1: Can you provide empirical evidence that unmodified, uncorrelated base models actually follow the uniform distribution for your random selection prompt? How does your p-value calculation account for inherent model biases toward specific letters?
> >
> > We acknowledge that certain models may have inherent biases toward specific letters, however, the binomial test in Eq. 5 is still valid and correct under these conditions. Please refer to our response to W2 and Appendix H.1 for a detailed proof.
> >
> > > Q2: Why were standard fingerprinting baselines (like Xu et al., 2024, which you cited) not considered from the empirical comparisons? How does RSP compare to these methods in terms of detection robustness under fine-tuning?
> >
> > We currently do not include direct comparisons primarily because it is extremely challenging to establish a fair evaluation with existing baselines. This difficulty arises from substantial differences in experimental settings across four critical dimensions:
> >
> > 1. Model types. The downstream models considered in prior work can differ significantly from ours. For example, ModelDiff is designed for image classification models, whereas our study focuses on LLMs and VLMs.
> >
> > 2. Data requirements. Our method requires no dataset, as the prefixes are randomly initialized. Whereas most existing approaches rely on curated datasets, which substantially affect performance and introduce dataset-dependent variability.
> >
> > 3. Model access. Our method does not require access to model weights or architectural details, while many baselines require partial or full access and, in some cases, even fine-tuning.
> >
> > 4. Evaluation metrics. Prior methods mostly output only a similarity score, and determining whether two models are correlated requires choosing an empirical threshold, raising concerns regarding its validity and generality. A key contribution of our work is that we eliminate such empirical design choices by introducing a hypothesis-testing framework that provides statistically rigorous p-values, making the correlation evidence significantly more reliable.
> >
> > Since we are the first to formulate model-correlation detection within a statistical hypothesis-testing framework, and because existing methods were not designed to produce statistically grounded p-values, there is no straightforward, principled metric that enables a fair comparison.
> >
> > To be specific, Xu et al. 2024 assumes access to and performs instruction fine-tuning on the model, which is fundamentally different from our setting. Regarding robustness under fine-tuning, our method already achieves 100% accuracy on the evaluated model pairs. More importantly, the main quantity we care about is the p-value, which these baselines do not provide.

---

> > > ### Author Response · Authors · 2025-11-17
> > > **Response to Reviewer 8XZy (3/3)**
> > >
> > > > Q3: Given the ~500 GPU-hour requirement for VLM prefix preparation in gray-box settings, is this method practically viable for real-world auditing? Have you explored if fewer prefixes (e.g.,
> > > ) suffice for VLMs given the very strong signal shown in Table 6?
> > >
> > >
> > > We argue that our method is practically viable for real-world auditing. Please refer to the response to W3 for detailed reasons. Additionally, we provide results for $K=50$, showing that our method can work with fewer prefixes.
> > >
> > > |               | Sample Number |  Gray-Box |  Gray-Box | Black-Box | Black-Box |
> > > |---------------|:-------------:|:---------:|:---------:|:---------:|:---------:|
> > > |               |               |  Visual7w | MathV360k |  Visual7w | MathV360k |
> > > | Qwen2.5-VL-7B |      500      | 1.00e-300 | 3.02e-208 | 1.00e-300 | 1.83e-205 |
> > > | Qwen2.5-VL-7B |       50      |  1.72e-49 |  2.83e-25 |  1.61e-45 |  2.83e-25 |
> > >
> > >
> > > Thank you again for your feedback. We are more than happy to discuss further if you have any additional questions or concerns.

---

> > > > ### Author Response · Authors · 2025-11-26
> > > > **Kind Reminder**
> > > >
> > > > This is a gentle reminder that the rebuttal period is approaching its deadline. If you have time, we would greatly appreciate it if you could review our responses and let us know whether they address your concerns.
> > > >
> > > > Thank you very much again for your time and thoughtful feedback!

---

### Official Review · Reviewer_LZaq · 2025-11-09

**Soundness:** 2
**Presentation:** 3
**Contribution:** 2
**Rating:** 4
**Confidence:** 4

**Summary:**

This paper introduces Random Selection Probing (RSP), a statistical hypothesis-testing framework for detecting model correlation. Unlike existing similarity-based methods that produce heuristic scores, RSP provides p-values by optimizing prefixes on a reference model for a random selection task (e.g., "choose a letter from a to z") and evaluating their transferability to a target model. The method works across LLMs and VLMs under diverse access conditions (gradient/logits-accessible for reference models; gray-box/black-box for test models). Experiments demonstrate extremely small p-values for fine-tuned models while maintaining high p-values for unrelated ones.

**Strengths:**

1. The problem of IP protection is urgent in the LLM era.

2. General Framework Design: RSP's two-stage pipeline (prefix optimization; correlation detection) considers various realistic scenarios. It uses multiple optimization techniques (GCG, GA, PGD, ZOO), and successfully handles:
- Model types: Both LLMs and VLMs;
- Access levels: Gradient or logits access during optimization, and grey-box or black-box settings during detection.

3. Robust False Positive Control: The introduction of an unrelated model $M_u$ to penalize generic features during optimization (maximizing $P_{M_r} - P_{M_u}$) is effective.

**Weaknesses:**

1. Opaque Transfer Mechanism: The paper relies on the key assumption of adversarial transferability and demonstrates that prefixes transfer but doesn't deeply analyze *why*. Key questions remain to answer: What specific "model-specific features" are encoded? Are they architectural artifacts, training data signatures, or optimization quirks?

2. Broad Applicability: How robust is transfer to more intense modifications other than fine-tuning (e.g., pruning, quantization, model merging, distillation, MoE upcycling)?

3. Hypothesis Test Model: The probability of binomial distribution is set as $1/N$. However, LLMs may have inherent tendencies towards some choices, which is not considered.

4. Adversarial Scenario: Such input optimization method is susceptible to detections and brittle to perturbations, which may not work well for test models with adversarial defense mechanism.

5. Computational Efficiency: While prefix optimization is a one-time cost, it can be substantial with many prefixes in need. This limitation could hinder adoption for resource-constrained auditing scenarios.

6. Lack of Related Work and Baseline Comparisons: Some works [1,2] also uses GCG optimization for model identification, thus should be mentioned and compared. No empirical comparison to existing model similarity methods (e.g., REEF, TRAP, LLMmap) is presented.

7. A minor issue: Two works share the name of AutoDAN [3,4], while the citation in Related Work might be mistaken according to the description.

[1] Gubri, Martin, et al. "Trap: Targeted random adversarial prompt honeypot for black-box identification." arXiv preprint arXiv:2402.12991 (2024).

[2] Tsai, Yun-Yun, et al. "RoFL: Robust Fingerprinting of Language Models." arXiv preprint arXiv:2505.12682 (2025).

[3] Liu, Xiaogeng, et al. "Autodan: Generating stealthy jailbreak prompts on aligned large language models." arXiv preprint arXiv:2310.04451 (2023).

[4] Zhu, Sicheng, et al. "Autodan: interpretable gradient-based adversarial attacks on large language models." arXiv preprint arXiv:2310.15140 (2023).

**Questions:**

1. The choice of candidate set. Have you tried other tasks other than choosing letters?

---

> ### Author Response · Authors · 2025-11-17
> **Response to Reviewer LZaq (1/3)**
>
> We sincerely appreciate the time and effort you have invested in reviewing our work, and thank you for your constructive feedback. However, we believe there are some misunderstandings regarding our paper. We would like to first clarify that our hypothesis test remains valid even in the presence of inherent biases in LLMs. Below, we address each of your concerns in detail.
>
>
>
> > W1 (a) Opaque Transfer Mechanism: The paper relies on the key assumption of adversarial transferability and demonstrates that prefixes transfer but doesn't deeply analyze why.
>
> Thank you for providing this interesting perspective on why adversarial transferability works. Adversarial transferability has been widely observed in prior work [1,2,3], and its underlying cause is a complex and still open research problem. However, this is beyond the primary scope of our paper. Our work focuses specifically on model correlation detection, and we leverage adversarial transferability merely as a tool to construct our detection method. A deeper investigation into the mechanisms behind adversarial transferability is indeed valuable, but we believe it is better suited for a separate, dedicated study.
>
> [1] Xie, Cihang, et al. "Improving transferability of adversarial examples with input diversity." Proceedings of the IEEE/CVF conference on computer vision and pattern recognition. 2019.
> [2] Zhu, Yao, Jiacheng Sun, and Zhenguo Li. "Rethinking adversarial transferability from a data distribution perspective." International Conference on Learning Representations. 2021.
> [3] Wang, Xiaosen, et al. "Admix: Enhancing the transferability of adversarial attacks." Proceedings of the IEEE/CVF international conference on computer vision. 2021.
>
>
> > W1 (b) Key questions remain to answer: What specific "model-specific features" are encoded? Are they architectural artifacts, training data signatures, or optimization quirks?
>
> Adversarial transferability can arise from multiple factors, including similarities in architecture, weight initialization, training data, optimization algorithms, and hyperparameters. All of these similarities may contribute to underlying model correlations. Importantly, our method is designed to capture such correlations effectively. As demonstrated in Tables 1–2, our approach successfully detects correlations induced by fine-tuning lineage, and in Tables 3–4, it also captures correlations in model families. Additionally, as noted in our response to W2, our method can also effectively capture correlations introduced by quantization and pruning.
>
>
>
> > W2: Broad Applicability: How robust is transfer to more intense modifications other than fine-tuning (e.g., pruning, quantization, model merging, distillation, MoE upcycling)?
>
>
> Below, we provide results on unstructured pruning and quantization for the Phi-4-mini-instruct model. The results show that our method remains robust under 4-bit quantization and retains its effectiveness with pruning ratios up to 50%.
>
>
>
> |        |      Gray-Box     |   Gray-Box  |   Gray-Box  |     Black-Box     |  Black-Box  |  Black-Box  |
> |--------|:-----------------:|:-----------:|:-----------:|:-----------------:|:-----------:|:-----------:|
> |        | 4bit Quantization | 30% Pruning | 50% Pruning | 4bit Quantization | 30% Pruning | 50% Pruning |
> |  Grad  |       1e-300      |   1.45e-76  |   3.03e-4   |       1e-300      |   1.53e-81  |   3.03e-4   |
> | Logits |       1e-300      |  9.08e-106  |   3.99e-3   |       1e-300      |  3.39e-110  |   3.99e-3   |

---

> > ### Author Response · Authors · 2025-11-17
> > **Response to Reviewer LZaq (2/3)**
> >
> > > W3: Hypothesis Test Model: The probability of binomial distribution is set as $\frac{1}{N}$. However, LLMs may have inherent tendencies towards some choices, which is not considered.
> >
> >
> > Thank you for raising the concern about potential assumptions on model bias. We would like to clarify that **our hypothesis test does NOT assume that the test model $M_t$ is unbiased over the candidate tokens**. In particular, we do **NOT** require $\mathbb{E}\_{ x \sim V^L} \mathbf{1}\left( \text{argmax}\_{j \in \\{1,\ldots, N\\}} P\_{M_t}(j \mid  x,p_r) = i \right) = \frac{1}{N}$
> > for any $i$. These expectations can be non-uniform and reflect arbitrary biases of $M_t$ toward specific tokens.
> >
> > The key idea is that under the null hypothesis $H_0$, (i) the optimized prefixes do not transfer from the reference model $M_r$ to the test model $M_t$, and (ii) the target index $i$ is sampled uniformly from $\\{1,\ldots,N\\}$ and is independent of the internal biases of $M_t$.
> >
> > Formally, let $a(\mathbf{x}, i, M_t, p_r) := \mathbf{1}\left(\text{argmax}\_{j} P_{M_t}(j \mid \mathbf{x}, p_r) = i\right) \in \\{0,1\\}$, and define $p_i :=\mathbb{E}\_{\mathbf{x} \sim V^L} [a(\mathbf{x}, i, M_t, p_r)]$. These $p_i$ capture exactly the original, possibly skewed, biases of $M_t$ toward each candidate token. We make **no assumption** that $p_i = 1/N$. The only structural property we use is that for any fixed $\mathbf{x}$, $\sum_{i=1}^N a(\mathbf{x}, i, M_t, p_r) = 1$, since exactly one token attains the argmax, taking expectations gives $\sum_{i=1}^N p_i = 1\$.
> >
> > Under $H_0$, the optimized prefix  $\mathbf{x}'$ behaves like a random prefix from the perspective of $M_t$, so $\mathbb{E}\_{\mathbf{x}}[a(\mathbf{x}', i, M_t, p_r)] = p\_i$ still holds. Because we draw $i \sim \mathrm{Unif}\\{1,\ldots,N\\}$, the success probability for a single trial is:
> >
> > $Pr(\text{success}) = \mathbb{E}\_i  \mathbb{E}\_{\mathbf{x}}[a(\mathbf{x}', i, M_t, p_r)]= \frac{1}{N} \sum_{i=1}^N p\_i= \frac{1}{N},$
> >
> > even though the individual $p_i$ can be arbitrarily biased. Intuitively, we are not testing “which token $M_t$ prefers,” but rather “whether $M_t$ outputs a *randomly designated* target token”; because the target is chosen uniformly and independently of $M_t$’s fixed preferences, the success probability is always $1/N$.
> >
> > We have updated the theorem and its proof in our paper (Appendix H.1) to make this point explicit: **the binomial model $X \sim B(K, 1/N)$ under $H_0$ holds without any assumption of uniformity in the test model’s token biases.**
> >
> > > W4: Adversarial Scenario: Such input optimization method is susceptible to detections and brittle to perturbations, which may not work well for test models with adversarial defense mechanism.
> >
> > Thank you for raising this concern. We are aware of this potential issue, which is part of why our prefixes are generated as random noise (random text for LLMs and random pixels for VLMs). Such random prefixes are unlikely to be specifically targeted by any adversarial defense mechanism. Moreover, to the best of our knowledge, there are currently no widely adopted adversarial defense techniques deployed in production-level models such as the Llama or Qwen series. This is largely because these defenses often introduce additional computation, increase inference latency, or lead to slight performance degradation on benign queries. For these reasons, we believe our method remains practical and effective for real-world applications.
> >
> > > W5: Computational Efficiency: While prefix optimization is a one-time cost, it can be substantial with many prefixes in need. This limitation could hinder adoption for resource-constrained auditing scenarios.
> >
> > We would like to clarify that the time cost of our method is already reported in Table 19. Although the logits-access setting for VLMs is relatively slow, in real applications one can simply use the LLM-side method, as modern VLMs also accept purely textual input. Moreover, as shown in Figure 2, our method remains effective with as few as 100 samples. Based on these observations, we summarize the estimated running time below:
> >
> > |     |        |  Per Sample  |   100 Samples  | 100 Samples on 8 GPUs |
> > |-----|--------|:------------:|:--------------:|:---------------------:|
> > | LLM | Grad   |  $\sim$ 65 s | $\sim$ 110 min |     $\sim$ 14 min     |
> > | LLM | Logits | $\sim$ 3 min | $\sim$ 300 min |     $\sim$ 38 min     |
> > | VLM | Grad   |  $\sim$ 23 s |  $\sim$ 40 min |      $\sim$ 5 min     |
> > | VLM | Logits | $\sim$ 3 min | $\sim$ 300 min |     $\sim$ 38 min     |
> >
> > Therefore, even in the worst case, our method completes within roughly 40 minutes on a standard 8-GPU compute node, which we believe is a reasonable cost for this task.
> >
> > Finally, we emphasize that determining model correlation is a high-stakes procedure in real-world IP protection, often involving dedicated teams. Given the seriousness of such investigations, allocating moderate computational resources is both standard and entirely justified.

---

> > > ### Author Response · Authors · 2025-11-17
> > > **Response to Reviewer LZaq (3/3)**
> > >
> > > > W6: Lack of Related Work and Baseline Comparisons: Some works [1,2] also uses GCG optimization for model identification, thus should be mentioned and compared. No empirical comparison to existing model similarity methods (e.g., REEF, TRAP, LLMmap) is presented.
> > >
> > > We currently do not include direct comparisons primarily because it is extremely challenging to establish a fair evaluation with existing baselines. This difficulty arises from substantial differences in experimental settings across four critical dimensions:
> > >
> > > 1. Model types. The downstream models considered in prior work can differ significantly from ours. For example, ModelDiff is designed for image classification models, whereas our study focuses on LLMs and VLMs.
> > >
> > > 2. Data requirements. Our method requires no dataset, as the prefixes are randomly initialized. Whereas most existing approaches rely on curated datasets, which substantially affect performance and introduce dataset-dependent variability.
> > >
> > > 3. Model access. Our method does not require access to model weights or architectural details, while many baselines require partial or full access and, in some cases, even fine-tuning.
> > >
> > > 4. Evaluation metrics. Prior methods mostly output only a similarity score, and determining whether two models are correlated requires choosing an empirical threshold, raising concerns regarding its validity and generality. A key contribution of our work is that we eliminate such empirical design choices by introducing a hypothesis-testing framework that provides statistically rigorous p-values, making the correlation evidence significantly more reliable.
> > >
> > > Since we are the first to formulate model-correlation detection within a statistical hypothesis-testing framework, and because existing methods were not designed to produce statistically grounded p-values, there is no straightforward, principled metric that enables a fair comparison.
> > >
> > > Regarding comparisons with the methods suggested by the reviewer, we note that REEF requires access to hidden states or activations, while our setting explicitly assumes no access to model internals. In addition, REEF only outputs an empirical similarity score without providing a statistical decision rule. RoFL also only outputs empirical TPR. TRAP and LLMmap focus on model fingerprinting or model identification, aiming to recover the exact model identity, which cannot measure model correlation, and therefore are not applicable to our problem setting. Our method also already achieves 100% accuracy on the evaluated model pairs.
> > >
> > >
> > > > W7: A minor issue: Two works share the name of AutoDAN [3,4], while the citation in Related Work might be mistaken according to the description.
> > >
> > > We apologize for the mistake. The citation has been corrected in the revision.
> > >
> > > > Q1: The choice of candidate set. Have you tried other tasks other than choosing letters?
> > >
> > > Our method is generally robust to the choice of the candidate set. Below, we present experiments using digits 0–9 as the candidate set on the Phi-4-mini-instruct model. The results show that our method continues to produce small p-values under this setting.
> > >
> > >
> > > |        | Gray-Box |  Gray-Box | Gray-Box | Black-Box | Black-Box | Black-Box |
> > > |--------|:--------:|:---------:|:--------:|:---------:|:---------:|:---------:|
> > > |        |   GSM8K  | Dolly-15k |  Alpaca  |   GSM8K   | Dolly-15k |   Alpaca  |
> > > |  Grad  |  1e-300  |  4.08e-3  | 2.00e-96 |   1e-300  |  4.08e-3  |  1.56e-95 |
> > > | Logits |  1e-300  |  1.77e-3  | 2.38e-67 |   1e-300  |  7.27e-4  |  1.40e-60 |
> > >
> > >
> > >
> > > Thank you again for your feedback. We are more than happy to discuss further if you have any additional questions or concerns.

---

> > > > ### Author Response · Authors · 2025-11-26
> > > > **Kind Reminder**
> > > >
> > > > This is a gentle reminder that the rebuttal period is approaching its deadline. If you have time, we would greatly appreciate it if you could review our responses and let us know whether they address your concerns.
> > > >
> > > > Thank you very much again for your time and thoughtful feedback!

---

### Meta-Review · Area_Chair_ixif · 2026-01-10

**Summary:**

The main concerns from the reviewers are following:

- Reviewer **LZaq**:
  - **W1**: The transfer estimation mechanism needs further clarification, such as the assumption of adversarial transferability and what specific model features are encoded.
  - **W2**: The robustness to broader model modifications beyond fine-tuning.
  - **W3**: The uniform assumption on output token distribution when doing hypothesis testing.
  - **W4**: The input optimization mechanism may be susceptible to detections and brittle to perturbations.
  - **W5**: Computational efficiency.
  - **W6**: Missing citations and baselines.

- Reviewer **8XZy**:
  - **W1**: The proposed approach may not be trustworthy enough for serious real applications.
  - **W2**: The uniform assumption on output token distribution when doing hypothesis testing.
  - **W3**: Computational efficiency.

- Reviewer **7EWa**:
  - **W1**: No comparison to existing baselines.

- Reviewer **zkVT**:
  - **W1**: The writing of the current paper lacks clarify and rigor.
  - **W2**: Failing the statistical test does not mean fine-tuning correlation.

**Reviewer Concerns:**

- **Concerns addressed in the rebuttal:**
  - Additional results are provided to show that the method can work beyond fine-tuning. Furthermore, the author provides additional proof to show that the uniform assumption on output token distribution is valid for the hypothesis testing task introduced in the paper. Finally, in my view, the computational efficiency is not a major issue since the computation lies in the training stage, and the optimized prefix can be reused for all target testing models.

- **Concerns remained outstanding:**
  - I agree with **W2** of Reviewer **zkVT** that the soundness of the proposed hypothesis testing method is still not fully justified. Even though accepting the null hypothesis can show that the target model has no correlation to the reference model, rejecting it only shows that the models are correlated, but can not answer where this correlation comes from. The reason can be complicated, such as sharing a common subset of training data, not necessarily "one model is generated from the other". I think this also shows that **W1** of Reviewer **LZaq** is valid: the current paper lacks in-depth analysis on what kind of transferability that the proposed approach can measure. If this is hard to analyze in theory, thorough experiments can also be valid, while the current paper lacks this even with the additional results provided in the author responses. I also agree with **W1** of **8XZy** that the current method may not be trustworthy enough for serious real applications.

  - Even though the authors provide discussions on the difficulty to compare with existing methods on transferability estimation, such experimental comparisons are actually necessary to clearly justify the application scenarios that the proposed method can indeed be utilized.

**Reviewer Scores:**

As discussed in the Reviewer Concerns, some points are indeed addressed by the author responses. While in my view, the remaining concerns (**W2** of Reviewer **zkVT**, **W1, W6** of Reviewer **LZaq**, **W1** of **8XZy**, **W1** of Reviewer **7EWa**) also prevent the reviewers with negative evaluations to fully change their opinions.

---

### Decision · Program_Chairs · 2026-01-26

Reject